# Convergent evolution of H4K16ac-mediated dosage compensation in the ZW species *Artemia franciscana*

Frederic Zimmer[1], Annika Maria Fox[1], Qiaowei Pan[1,2], Frank Rühle[1], Peter Andersen[3], Ann-Kathrin Huylmans[4], Tanja Schwander[2], M. Felicia Basilicata[1,5], Claudia Isabelle Keller Valsecchi[1,6]*

1 Institute of Molecular Biology (IMB), Mainz, Germany, 2 Department of Ecology and Evolution, Université de Lausanne, Lausanne, Switzerland, 3 Department of Molecular Biology and Genetics, Aarhus University, Aarhus, Denmark, 4 Johannes Gutenberg-Universität Mainz, Mainz, Germany, 5 University Medical Center (UMC), Mainz, Germany, 6 Biozentrum, University of Basel, Basel, Switzerland

* c.kellervalsecchi@unibas.ch

## Abstract

Sex chromosomes impact chromatin organization and histone modification dynamics differently between males and females, particularly those involved in dosage compensation (DC). The evolutionary diversity, as well as the tissue- and age-dependent variations of DC mechanisms are incompletely understood. Here, we investigate the occurrence of histone H4 lysine 16 acetylation (H4K16ac), previously known for its role in sex chromosome DC in the male-heterogametic fruit fly *Drosophila melanogaster* and the green anole lizard *Anolis carolinensis*. By sampling multiple arthropods, we find the convergent evolution of H4K16ac for DC in a female-heterogametic (ZW) species, the crustacean *Artemia franciscana*. CUT&Tag analysis demonstrates that H4K16ac is confined to the non-recombining stratum of the Z chromosome in females. H4K16ac-mediated DC is established during embryogenesis. In aged individuals, we observe an overall decline in nuclear organization, disrupted H4K16ac territories and increased variability in local acetylation levels on the female Z chromosome. Our findings shed light on the evolutionary diversity of DC across species and raise the possibility of sex-specific histone acetylation contributing to male-female differences in lifespan.

## Author summary

Sex chromosome dosage compensation (DC) mechanisms restore gene expression balance in species with heteromorphic sex chromosomes, which occur either as male-heterogametic systems (XX/XY, with males carrying a single X) or female-heterogametic systems (ZZ/ZW, with females carrying a single Z). Across taxa, a recurring theme is the involvement of histone modifications in regulating

which permits unrestricted use, distribution, and reproduction in any medium, provided the original author and source are credited.

**Data availability statement:** The developmental transcriptome RNA-seq data have been deposited at GEO under the accession GSE289905. CUT&Tag data of D. melanogaster and Bacillus grandii are deposited at GEO under the accession GSE306499. All other datasets are deposited under PRJNA1197400.

**Funding:** CIKV is funded by the Deutsche Forschungsgemeinschaft (DFG, German Research Foundation, https://www.dfg.de/de) - Individual Grant Project no. 513744403, Scientific Network Grant Project no. 531902894, GRK2526 "Genevo" - Project no. 407023052", GRK2859 ("4R") - Project no. 491145305, Forschungsinitiative Rheinland-Pfalz (ReALity, https://reality.uni-mainz.de/), the EMBO Young Investigator Program (5795, https://www.embo.org/) and institutional funding from the Institute of Molecular Biology. MFB received financial support from the intramural High Potentials Grant program of the University Medical Center Mainz (https://www.unimedizin-mainz.de/index.html) and Forschungsinitiative Rheinland-Pfalz (ReALity). The funders had no role in study design, data collection and analysis, decision to publish, or preparation of the manuscript.

**Competing interests:** The authors have declared that no competing interests exist.

this balance at the chromatin level. One such modification, histone H4 lysine 16 acetylation (H4K16ac), is an activating mark well characterized for its role in DC in the XY model organism *Drosophila melanogaster*. To assess its broader evolutionary relevance, we examined H4K16ac across multiple arthropod species. Our analysis identified this histone mark as part of the DC mechanism in the ZW crustacean *Artemia franciscana*, expanding its function to female heterogametic species. We find that H4K16ac localizes specifically to the differentiated part of the single Z in females, promoting gene upregulation to match the expression levels of the two Z in males. This acetylation pattern for DC is established in early embryos and becomes increasingly variable in old adult females. Our study contributes to the growing understanding of the evolutionary diversity of DC and points to a potential role of these processes in sex-specific aging.

## Introduction

In many species, males and females display profound differences in morphology, physiology, behavior or lifespan. Sex is often determined by heteromorphic sex chromosomes, which evolved from autosomes and display remarkable evolutionary dynamics [1]. Besides the well-known XY/XX chromosomes, the ZW nomenclature refers to a configuration with heterogametic females (ZW) and homogametic males (ZZ). Unlike the XY system where sperm determines sex (e.g., in humans or *Drosophila*), in the ZW system (e.g., in birds, some fish and crustaceans), the sex of the offspring is determined by the ovum [1, 2]. Under both sex chromosome systems, the copy number of sex-chromosomal genes in the heterogametic sex is halved (ZW females and XY males).

Differences in copy numbers bear the potential to be detrimental, for example because gene regulatory networks might be disrupted, transcription noise may be increased and multisubunit complexes may become imbalanced [3]. Dosage compensation (DC) evolved in some species to re-equilibrate gene expression of sex-chromosomal genes between males and females [4, 5]. DC mechanisms typically manifest at the chromatin level and their study has been fundamental to our understanding of transcription, histone modification, three-dimensional genome architecture and non-coding RNAs [6]. However, most insights have been derived from XY systems. ZW systems remain poorly characterized, although hundreds, if not thousands of species exhibit a ZW configuration [1, 2]. Theoretical models centered on sexual selection differences between the Z and X chromosomes [7], along with transcriptome analysis in the "model" ZW organism, the chicken, initially led to the assumption that DC would be entirely absent in ZW systems [8] or, if present, would act only on few dosage-sensitive genes [9]. Recent evidence, however, suggests that chicken DC is achieved at posttranscriptional levels [10] via a molecular mechanism involving a male-essential miRNA [11]. In addition, lepidopteran insects such as *Danaus plexippus* (monarch butterfly) [12, 13], the branchiopod *Artemia franciscana* (brine shrimp) [14], the trematode parasite *Schistosoma mansoni* (blood

fluke) [15] and the reptile *Apalone spinifera* (spiny softshell turtle) [16] appeared more recently as notable exceptions that exhibit DC.

While these studies described the existence of DC, the molecular mechanisms providing equilibration in these ZW systems remain unclear. One study in monarch butterflies showed that the neo-Z chromosome, which arose from an evolutionary recent fusion between the ancestral Z (a chromosome conserved across Lepidoptera) and an autosome, is upregulated in ZW females and enriched for histone H4 lysine 16 acetylation (H4K16ac) [13]. In contrast, the ancestral Z undergoes dosage dampening in ZZ males by a so far unknown molecular mechanism [13]. In the well characterized XY model systems (mammals, the nematode *Caenorhabditis elegans*, the dipteran *Drosophila melanogaster*) different molecular mechanisms are found for DC [17]. Mammals and *C. elegans* use repressive chromatin modifications in XX individuals along with compensatory mechanisms that restore X-to-Autosome balance [18–20]. Male fruit flies upregulate the single X in order to match the expression to the XX females [21]. This is achieved via H4K16ac, deposited by the Male-specific lethal (MSL) complex, which induces chromatin decompaction [21]. H4K16ac does not mediate upregulation of the male X in another dipteran insect, the malaria mosquito *Anopheles gambiae* [22, 23]. Conversely, H4K16ac is enriched on the single X of male *Anolis* lizards [24]. Apparently, H4K16ac appears to play a crucial role in DC across taxonomically diverse species with independently evolved sex chromosomes. However, because the presence of H4K16ac for DC has only been investigated in single-organism studies, it is unknown how common H4K16ac-DC is, whether it occurs more widely in ZW species, and to what extent it can vary across tissues and life stages. Of note, H4K16ac is also known to control the expression of transposons [25], which play vital roles in evolutionary innovation and can show distinct dynamics on sex chromosomes [26].

Besides its role in X chromosome DC, H4K16ac is important to regulate developmental processes [27] and impacts lifespan. H4K16ac levels are increased in old yeast cells and diminishing H4K16ac levels increases lifespan [28]. Similarly, there is a redistribution of H4K16ac in old versus young human brains [29] and H4K16ac changes subnuclear polarity in aged hematopoietic stem cells [30]. The heterogametic sex (ZW females, XY males) is often more short lived than the homogametic sex [31]. For *Drosophila*, it has been proposed that heterochromatin loss on the Y upon aging can become toxic and shortens lifespan [32, 33], however, this view has been recently challenged [34]. While these observations suggest correlations between sex chromosome chromatin and its variation across life stages, it remains unclear whether H4K16ac changes across the lifespan and this contributes to modulation in DC.

Here we investigate the sex-specific patterns of H4K16ac across arthropods to probe its general involvement in DC. We find that this modification is not universally present, yet has been convergently co-opted for DC in an arthropod with a ZW chromosome system, the crustacean *A. franciscana*. In line with a recent report from [35] we show that H4K16ac is enriched in a *Drosophila*-like fashion on the differentiated Z chromosome. Curiously, the MOF-MSL complex shows no sex-specific expression patterns, pointing towards the possibility of different upstream regulatory factors for DC. Lastly, we characterize the H4K16ac-DC dynamics across lifespan and show that DC onset occurs during embryogenesis but then becomes variable during *Artemia* aging along with a global deterioration of chromatin organization.

## Results

### Sex-specific patterns of H4K16ac across arthropods

In male heterogametic insects, DC is often achieved through the upregulation of the X in males (e.g., the X chromosome in Diptera), while in the female heterogametic lepidopterans, it is achieved via upregulation of the single Z in females and partial downregulation of both Zs in males (Fig 1A) [5, 12, 36].

We investigated whether the independent evolution of heteromorphic sex chromosomes is associated with the same molecular mechanism of upregulation, specifically H4K16ac. To explore this, we selected seven insect species in which heteromorphic sex chromosomes evolved independently at least five times (indicated by colored bars for each independent

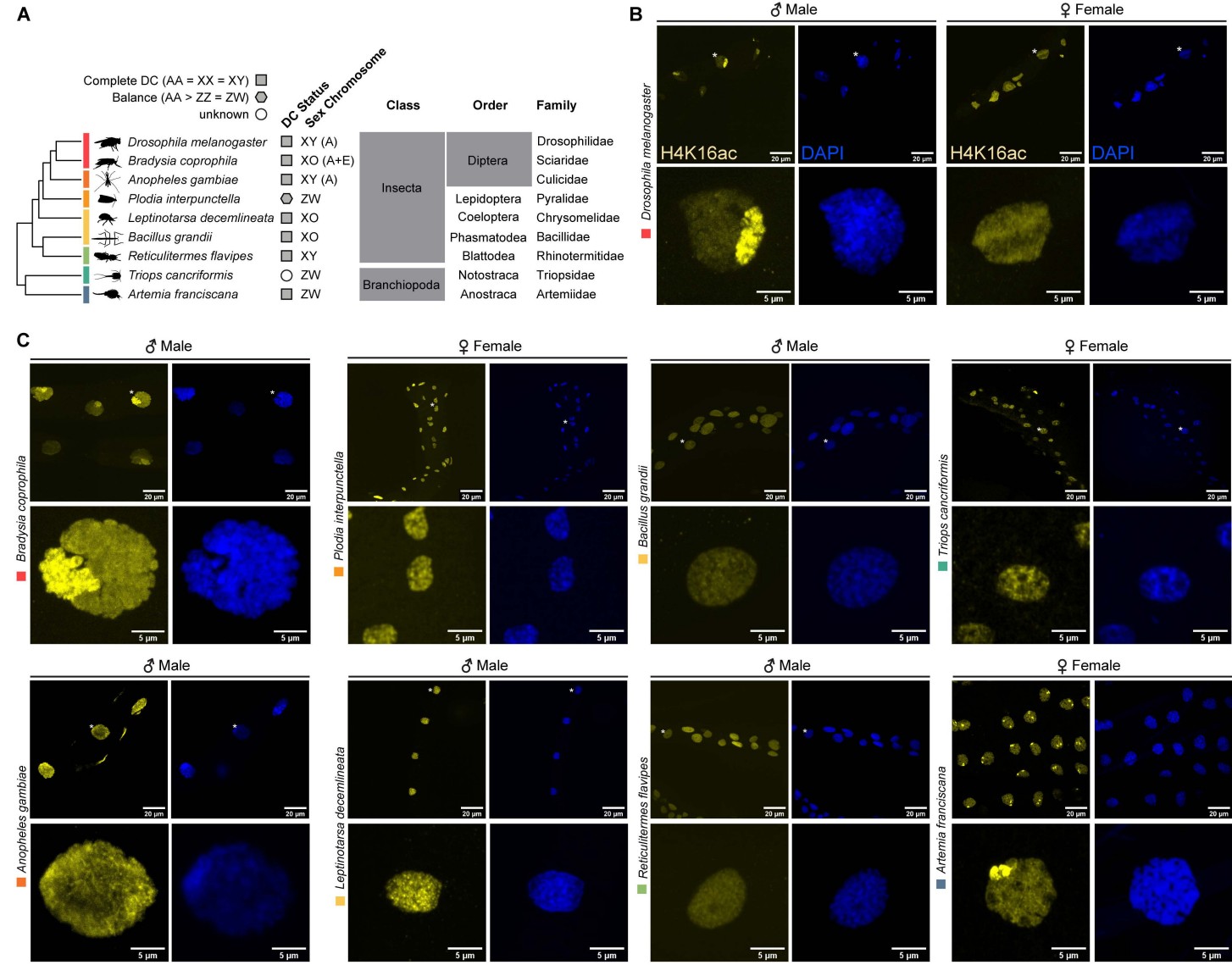

**Fig 1. Sex-specific enrichment of H4K16ac across arthropods.** (A) Schematic evolutionary tree and taxonomy of the analyzed arthropod species. Independent evolutionary events that gave rise to sex chromosomes are indicated with colored rectangles. Filled symbols indicate known (grey) or unknown (white) DC status, respectively. DC presence for the selected species in [12,14,37–41] and S1B Fig. Pictograms are from phylopic.org. (B) Immunofluorescence staining of H4K16ac in yellow and DAPI in blue of adult *D. melanogaster* males and females. All images represent maximum intensity projections of a Z-stack. (C) as in (B) but for the heterogametic sex of the indicated Arthropod species in adult tissues. The homogametic sex stained/acquired as part of the same experiment are shown in S1A and S1C Fig.

event; Fig 1A; [42]). We further broadened the taxonomic sampling by adding two female heterogametic crustaceans. Enrichment of H4K16ac was visualized through immunofluorescence staining followed by confocal microscopy. If H4K16ac mediates upregulation, we expected that the X/Z chromosome can be identified as a sex-specific subnuclear compartment with more intense staining compared to the nuclear background (referred to as a "territory", a hallmark of DC; Fig 1B). Confirming previous findings, we detected an H4K16ac-positive territory in male *D. melanogaster* (Fig 1B) and the same pattern was observed in the fungus gnat *Bradysia coprophila* (Figs 1C and S1A). This aligns with the fact that the

ancestral autosome, Muller Element (ME) A, became the X chromosome before these two dipteran species diverged [36]. This suggests that the H4K16ac-DC mechanism evolved once in a common ancestor of these species and then remained conserved. By contrast, as previously reported [22, 23], the independently evolved X chromosome of *Anopheles gambiae* (Diptera) - albeit originating from ME-A - uses a distinct DC mechanism that does not involve H4K16ac. Accordingly, no territory was observed, and the nuclear staining was uniform in both sexes (Figs 1C and S1A). Similarly, the Indian meal moth (*Plodia interpunctella*) did not show enrichment of the activating H4K16ac mark in either sex (Figs 1C and S1A). *Leptinotarsa decemlineata* (Coleoptera, beetles [43, 44]; S1B Fig for DC status) and *Bacillus grandii* (Phasmatodea, stick insects [39]) share a deeply conserved, homologous X chromosome content that predates the evolution of Insecta [42]. Neither males nor females in these two species exhibited an H4K16ac-positive territory (Figs 1C and S1A). The termite *Reticulitermes flavipes* offers another intriguing case to study DC. Within the order Blattodea, termites transitioned from the ancestral XO system to a newly evolved XY system with complete DC [40]. We found no H4K16ac-positive territory in male termites (Figs 1C and S1A), suggesting, again, an H4K16ac-independent mechanism for upregulation. These findings indicate that the ancestral DC mechanism in insects fundamentally differs from the *Drosophila* MSL-H4K16ac mechanism and may have been lost and replaced during sex chromosome turnover in different orders, such as Diptera. Lastly, we examined two arthropods outside Insecta: the brine shrimp (*A. franciscana*) and the European tadpole shrimp (*Triops cancriformis*), both of which exhibit female heterogamy (ZW) [14, 45]. Notably, we observed a female-specific H4K16ac-positive territory in *A. franciscana*, but not in *T. cancriformis* (Figs 1C and S1C). The H4K16ac-territories in female *Artemia* were present in all examined adult tissues (e.g., in epithelial cells of the abdomen and the rectum; or the head, e.g., in the antennae or eye stalks; S1D Fig). Males instead showed a homogeneous staining in the nucleus without focal enrichment in all examined tissues (S1C Fig). This indicates that DC occurs in all tissues of adult *Artemia*.

To validate that the presence or absence of a territory in microscopy can accurately capture sex chromosome enrichment of H4K16ac, we analyzed this mark by genome-wide methodology in four selected species: *A. gambiae* (no territory, data from [23]) along with new data generated in *B. grandii* (no territory), *D. melanogaster* (territory) and *A. franciscana* (territory). For the latter three, we performed CUT&Tag (Cleavage Under Targets and Tagmentation [46]) chromatin profiling on adults using H4K16ac antibodies and IgG controls. CUT&Tag is a highly sensitive technique similar to ChIP-seq, enabling precise identification of genomic regions enriched with a specific protein or histone modification (for mapping rates and quality control see S1 Data). After mapping CUT&Tag and ChIP-seq datasets, respectively, we segmented the autosomes and X/Z chromosomes into equally sized bins to analyze H4K16ac enrichment. There was a substantial sex-specific enrichment of H4K16ac on X of *D. melanogaster* (Figs 2A and S2A) and Z of *A. franciscana* (Fig 2B) but not the X of *B. grandii* nor *A. gambiae* (Fig 2A). When we compared the *B. grandii* H4K16ac enrichment with the positive control profile generated in parallel (RNA Polymerase II antibody) (S2A Fig), we found that H4K16ac was found at the transcription start sites of highly expressed genes (S2B Fig) - a pattern reminiscent of that observed in *Drosophila* females [27] and *Anopheles* [23]. This refutes a technical explanation for the dissimilarity of H4K16ac on the X of male stick insects and fruit flies, respectively, and indicates that these species likely achieve DC by two distinct mechanisms. Conversely, in *A. franciscana* the female Z chromosome exhibits significantly higher H4K16ac levels compared to the male Z or autosomes (Figs 2B and S2C; IgG controls in S2D Fig).

In summary, two independent evolutionary events of sex chromosome evolution, one resulting in male-heterogamety (in dipterans) and one resulting in female-heterogamety (*Artemia*), resulted in H4K16ac-mediated DC. Considering the lineage-specific evolution of DC in malaria mosquitoes (which rely on the *SOA* gene that is absent in other insects [22]), we conclude that the seven instances of sex chromosome evolution studied here involve at least four distinct DC mechanisms.

## H4K16ac is confined to the ancient *Artemia* Z chromosome stratum

Having confirmed that the territories (Fig 1) in female *Artemia* correspond to the Z chromosome, we were interested in investigating this system in further detail. The *A. franciscana* Z contains two evolutionary strata: a younger

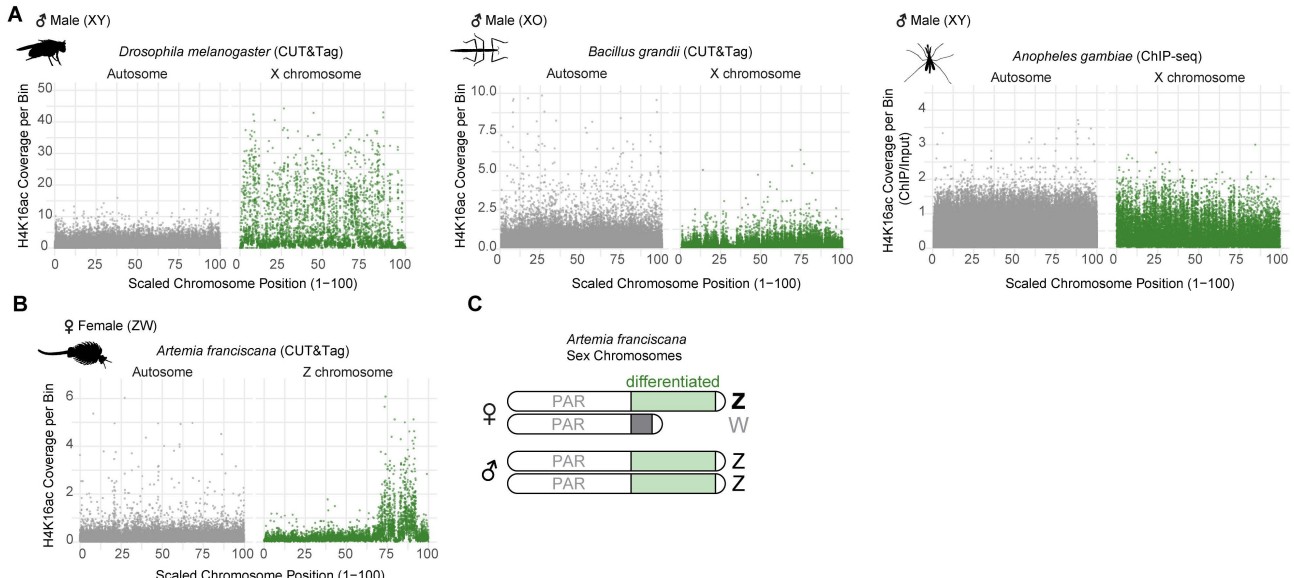

**Fig 2. Genome-wide profiling of H4K16ac in four Arthropod species.** (A) Dot plots showing the normalized H4K16ac CUT&Tag or ChIP-seq, respectively, coverage of a single representative replicate for autosomes and X chromosomes, segmented into 5 kb bins, in males of the indicated species. See also S1 Data for quality control and S2 Fig for females and positive controls. The x-axis represents the relative position of each 5 kb bin along the chromosome, scaled from 1 (chromosome start) to 100 (chromosome end). (B) as in (A) but for the H4K16ac CUT&Tag data in a single representative *A. franciscana* female with 10 kb bins (C) Scheme of *A. franciscana* sex chromosomes containing a pseudoautosomal region (PAR) as well an adjacent region, where Z and W are differentiated. Pictograms are from phylopic.org.

pseudoautosomal region (PAR), where the Z and W chromosomes remain highly homologous and undifferentiated due to ongoing recombination, and an older region, where the Z (referred to here as Z(diff)) and its corresponding segment on the W are differentiated [47] (Fig 2C). We inspected whether the H4K16ac signals differ in the two evolutionary strata, Z(diff) and PAR. For this, we divided the Z chromosome into 14 equally sized windows and found that the H4K16ac coverage was drastically higher in the Z(diff) region compared to the PAR (Fig 3A). Comparison of female and male coverage in the genome browser revealed that H4K16ac enrichment was highest in the evolutionary most ancient region (S0 [47], median log2FC female/male 1.280) and intermediate in the less differentiated S1 (median log2FC female/male 1.169) and S2 (median log2FC female/male 0.435) regions (Fig 3B). In the PAR, H4K16ac levels were similar between females and males (median log2FC 0.00), and the same was true for the autosomes (median log2FC 0.00; S2E Fig).

We then grouped the *A. franciscana* genes by chromosomal location (Z(diff), PAR and autosomes) and RNA expression status (expressed: ≥ 10 TPM; not expressed: < 10 TPM) and analyzed enrichment at transcription start sites (TSS) and within gene bodies (Fig 3C; IgG controls in S2D Fig). In males (Z chromosome and autosomes) and females (autosomes), the H4K16ac signal was restricted to the promoters of expressed genes, with no enrichment in gene bodies and at inactive genes (Fig 3C). In contrast, the female Z chromosome exhibited broad H4K16ac enrichment across both active and inactive genes, with the signal extending into intergenic regions (i.e., regions between genes). Active Z-linked genes showed higher H4K16ac levels compared to inactive ones and displayed a prominent peak around the TSS. Overall, this distribution (broad on sex chromosome in heterogametic sex; promoter on autosomes and in homogametic sex) closely resembles that observed in *Drosophila* [23]. Our findings are illustrated in the region shown in the genome browser snapshot (Fig 3D), where the broad spreading of H4K16ac beyond transcription units (visualized by RNA-seq) on the female Z(diff) is clearly visible, while in males, the signal remains restricted to regions around TSS.

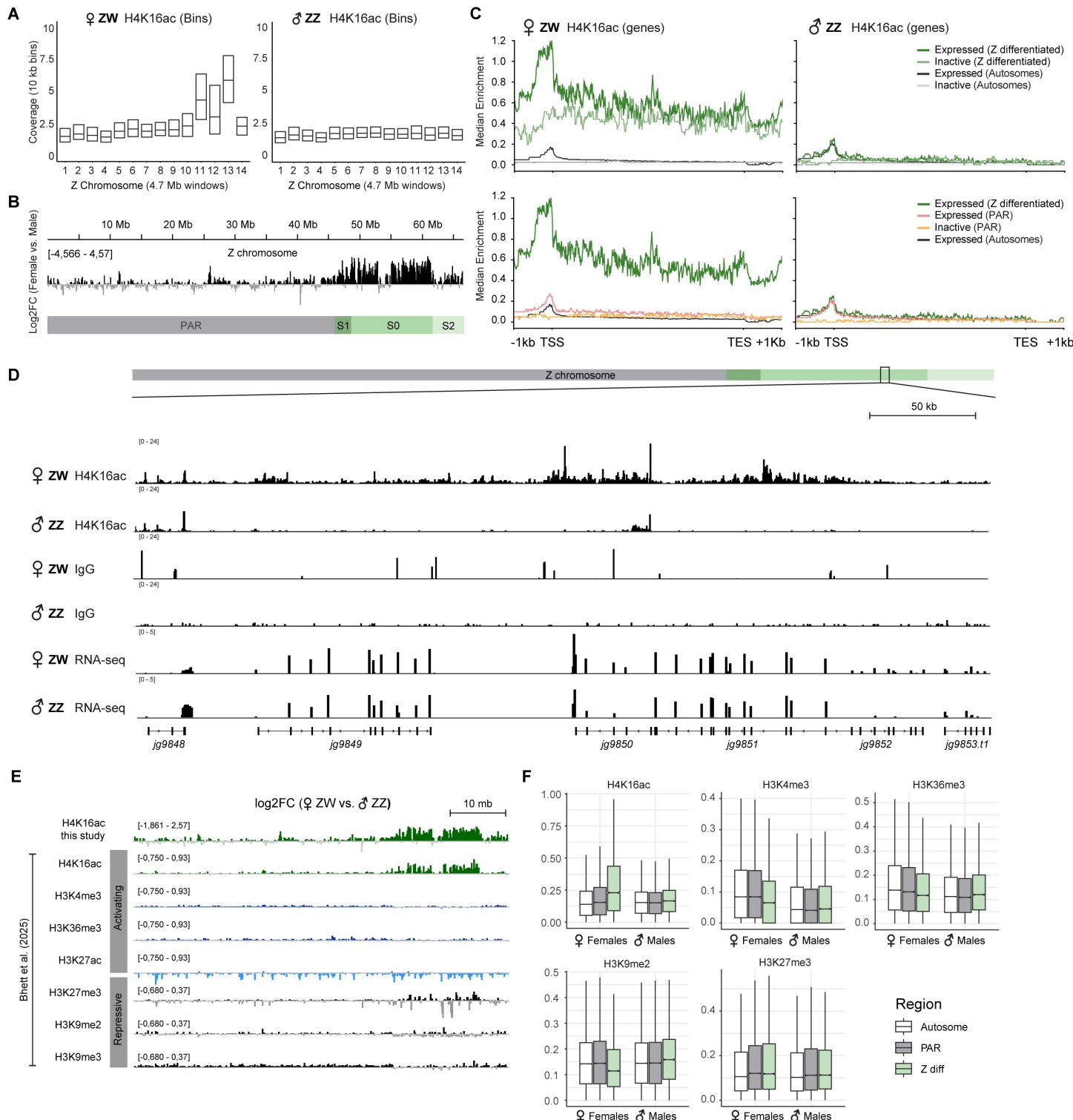

**Fig 3. H4K16ac enrichment on the female *A. franciscana* Z chromosome.** (A) Box plots showing the count per million (CPM) normalized read counts for the Z chromosome segmented into 10kb bins in females (left) and males (right) from H4K16ac CUT&Tag profiles. 14 equally sized windows of 4.7 Mb along the Z were generated and the distribution of enrichment in the 10 kb bins shown as boxplots. (B) log2FC of H4K16ac levels between females and males across the Z chromosome, highlighting enrichment in the Z(diff) region and S1, S0 and S2 strata as defined in [47]. (C) CPM-normalized H4K16ac CUT&Tag enrichment of

a single representative female (left) and male (right) replicate shown as a metaplot with the median enrichment across different gene groups. The gene bodies were scaled to 5kb between the TSS and transcription end site (TES), while extending by 1kb up- and downstream, respectively, for plotting. Genes were classified as expressed based on a cutoff of TPM ≥ 10 and further distinguished based on chromosomal location (Autosomes, PAR and Z(diff)). The curves in top and bottom plots for Z(diff) and expressed (Autosomes) are the same to enable comparisons among the different regions and gene groups. IgG controls and further replicates are shown in S2D Fig. (D) Genome browser snapshots of CPM-normalized H4K16ac or IgG CUT&Tag coverage as well as RNA-seq of a representative region in the Z(diff) in males and females. The RNA-seq data (normalized coverage) is from [14]. (E) log2FC of indicated histone modification levels between females and males across the Z chromosome. Data from this study and [35]. (F) Boxplots showing the CPM-normalized histone modification CUT&Tag coverage for each chromosomal region (autosomes, PAR and Z(diff)) segmented into 10kb bins, in females and males.

To get further insights into the chromatin landscape of the dosage-compensated Z, we reanalyzed histone modification profiling data from [35] by comparing the female and male signals as log2 fold change (log2FC) in the genome browser. In line with what had been reported by [35], this showed that different from H4K16ac, no other marks show such a striking pattern on the Z (Fig 3E). Using boxplots (Fig 3F), we inspected this more closely and found that H4K16ac is the only activating mark that is highly enriched on the Z(diff) (Wilcoxon $p$-value 1.25E-188 Z(diff) vs. autosomes). Interestingly, there was a minor depletion for H3K4me3 (active; $p = 4.19E-40$) and H3K36me3 (active; $p = 5.62E-14$), as well as H3K9me2 (repressive; $p = 2.90E-24$) on the female Z(diff) compared to autosomes or PAR that was not visible in males. Those could contribute to fine-tuning necessary to achieve the approximately 2-fold effect for DC.

### Absence of sex-specific patterns of MOF-MSL complex in *Artemia*

H4K16ac on the male X chromosome in *Drosophila* is deposited by the MSL complex (S3A Fig). Accordingly, three (*msl-2*, *msl-3* and *mof*) out of four core MSL complex subunits show significantly male-biased expression (S3B Fig) (Mean normalized read counts in females (F) and males (M): *msl-2* F 408, M 1645; *msl-3* F 448, M 821; *mof* F 311, M 1084). Like H4K16ac, MSL proteins localize to a male-specific territory (shown for MOF in S3A Fig). This distinguishes the MSL members from other histone acetylation factors such as *Gcn5*, which show no male-biased RNA expression in fruit flies (S3B Fig). In species like *A. gambiae*, where the MSL complex has no function in DC, MSL core subunits lack sex-specific expression (S3B Fig). As there is no sex-specific H4K16ac territory (Fig 1C), we did not expect sex-biased expression of MSL subunits in *L. decemlineata*, and analysis of a sex-specific expression atlas dataset [48] indeed confirmed this expectation (S3B Fig).

Through BLAST searches of structured domains of *Drosophila* MSL subunits (e.g., MOF HAT or MSL2 RING) we identified orthologues of MSL proteins in *A. franciscana*. In RNA-seq data of adults, none of the *A. franciscana* MSL orthologues showed a female-biased expression (S3B Fig). Because DC regulators such as MSL2 and SOA are also controlled through co- and post-transcriptional mechanisms, including sex-specific splicing [22, 49, 50], we examined the splicing patterns of all *Artemia msl* orthologues, but found no evidence of sex-specific events like, e.g., intron retention or exon skipping (S3C Fig). In addition, sequence and structural alignments [51] of *Drosophila* with *Artemia* MSLs revealed high conservation of the structured domains, e.g., in the MOF histone acetyltransferase domain or the MSL3 MRG domain (S4A-S4D Fig). However, *Artemia* MSLs lack critical protein extensions that specifically evolved for a function in DC in *Drosophila*, such as the MOF N-terminus [52] or the MSL2 C-terminal domain [53].

This suggests the existence of another sex-specific component of the MOF-MSL complex or alternatively, that *Artemia* may employ an entirely different upstream regulatory mechanism that induces H4K16ac. Given recent evidence for enrichment of an AT-rich motif on the Z(diff) [35] - previously identified as a binding site for Human antigen R (HuR) - we examined HuR orthologues as potential sex-specific components of the *Artemia* DC mechanism. However, none showed markedly higher expression in females; in fact, one was more highly expressed in males (S3B Fig).

### *Artemia* DC is established in the embryonic stage

Previous studies in *Artemia* investigated the presence of DC in adults using bulk RNA-seq [14] and tracked Z chromosome expression dynamics in the female germline [54]. However, the onset and stability of DC across life stages are

PLOS Genetics

currently unknown. Notably, all well-understood DC mechanisms share a common feature: their establishment early in embryogenesis [18, 22, 55–57]. In *Artemia*, embryonic development lasts approximately five days, ending with the hatching of free-swimming nauplii (Fig 4A). This is followed by a juvenile stage during which the sexes are immature and visually indistinguishable. Sexually mature adults with visible male-female differences emerge at around 25 days post fertilization (dpf).

To investigate the onset of DC, we performed low-input RNA-seq on single embryos at 1–2 days post-fertilization (dpf) and 4 dpf (mapping rates and quality control in S1 Data). Those stages correspond to blastula (1–2 dpf) and post-gastrulation (4 dpf) stages, respectively [58]. To confirm that we sampled distinct developmental stages we performed euclidean distance analysis showing that the two time-points clustered separately (S5A Fig) and differential gene expression analysis identified 739 up- and 445 downregulated genes (4 dpf vs. 1–2 dpf; S5B Fig). In addition, we examined the expression of *Artemia* orthologues of key early developmental regulatory genes in *Drosophila*. Observing that the pair-rule gene *fushi tarazu* (*FTZ*), head gap gene *empty spiracles* (*EMX2*), *Nicotinic Acetylcholine Receptor Subunit b1* (*nAChRb1*, nervous system development) and the transcription factor *PAX5* (midbrain development) are expressed at the 1–2 dpf time-point indicates that embryos had passed zygotic genome activation, while their expression dynamics confirms that the 4 dpf embryos undergo tissue specification and organogenesis (post-gastrulation) (S5C Fig). *RPL19* - a housekeeping gene - appeared stable, while the homeobox gene *UBX* - whose function may have diverged in Crustacea [59] - was not expressed.

Next, we determined the sex of the embryos based on single nucleotide polymorphisms (SNPs) in transcripts from the Z(diff) region, where females are expected to feature significantly reduced heterozygosity as compared to males (see Methods, S2 Data and S5D-S5F Fig). We then performed differential gene expression analysis of ZW females in comparison with ZZ males for the two time-points. This showed that at 1–2 dpf, the Z(diff) genes overall show significantly lower expression in females compared to males (Fig 4B). No differences could be scored at the later 4 dpf stage (Fig 4B). This was also reflected by a comparison of 4 dpf versus 1–2 dpf females, where there was a clear induction of gene expression for the Z(diff) genes, but no global effect for genes on PAR or autosomes (Fig 4C). This indicates that DC establishes during embryogenesis and is complete before the hatching of nauplii larvae. To independently validate this finding, we performed immunostaining of H4K16ac to pin-point the timing of sex chromosome territory appearance during embryo development. In agreement with the RNA-seq data, 1 dpf embryos exhibited a uniform H4K16ac staining pattern, with all embryos (assuming a 50:50 male-to-female ratio) showing homogeneously stained nuclei (Fig 4D). In contrast, by 3 dpf, roughly half of the embryos (which we assume to be females) displayed a Z chromosome-territory, while the other half (likely ZZ males) showed homogeneously stained nuclei (Fig 4E).

We explored this dataset for putative upstream regulators inducing H4K16ac, but did not find a female-specific induction of any MSL subunits (Fig 4F) or other histone acetylation factors (S5G Fig) during embryogenesis. We also searched for putative *cis*-linked non-coding RNAs analogous to roX1/2 (fruit flies), Xist (mammals), or MAYEX (lizards). Using the binning approach of [60] that does not rely on prior annotation of a given transcript, we identified sex-biased candidate regions (S5H Fig); however, the most significant hits mapped to autosomes, and the few on Z were artifacts caused by homology to the W. Manual inspection of the remaining top hits revealed that they were either lowly or not expressed at all in adults (S3 Data). We also applied IRFinder [61] to detect sex-biased intron retention events (S4 Data), but the five significant hits (padj<0.05) were all false positives when inspected manually (S3 Data). In summary, our analyses were unable to identify a strong candidate for an upstream regulatory factor that promotes female-specific H4K16ac on Z(diff). Although these results do not exclude the possibility that acetylation is deposited by MOF-MSL like in *Drosophila*, we find it interesting to observe key differences in sex-specific regulation in the *Artemia* system. Besides MOF-MSL, another speculative possibility is that H4K16ac enrichment on the female Z involves a distinct acetyltransferase complex or differential regulation of an eraser complex (see Discussion).

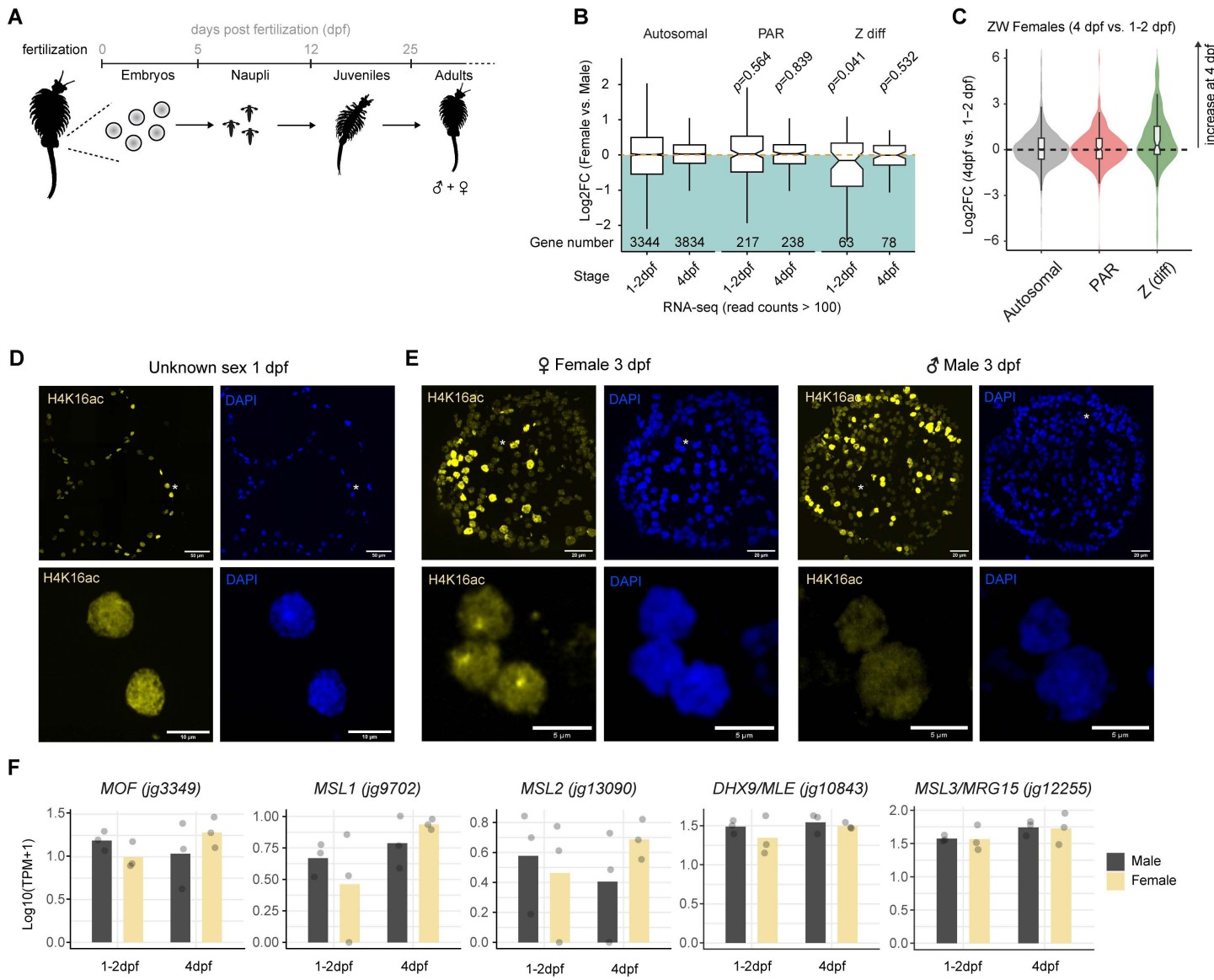

**Fig 4. Onset of H4K16ac-mediated DC during embryogenesis.** (A) Schematic Illustration of *A. franciscana* development and life stages in days after fertilization (dpf). Pictograms are from phylopic.org. (B) Boxplot showing expression changes between females and males by low-input RNA-seq conducted from single 1-2 and 4 dpf embryos. The log2FC for genes were determined with DESeq2 (female vs males) for each time-point (*n*=3 embryos for each sex and time-point). The log2FC values are plotted for autosomal, PAR and Z(diff) genes that are expressed (> 100 average read counts). The gene numbers shown in each boxplot are listed below. The *p*-values were determined using a one-sided Wilcoxon signed rank test with continuity correction. (C) RNA-seq as in (B) but comparing female 4 dpf with female 1-2 dpf samples. The log2FC distribution of the expressed genes in the different chromosomal locations is shown as a violin plot with an overlaid boxplot. (D) Immunofluorescence staining of H4K16ac in yellow and DAPI in blue of cryosections of *A. franciscana* embryos at 1 dpf (*n*=39 embryos were analyzed). (E) as in (D) but for *A. franciscana* embryos at 3 dpf (*n*=25 embryos). The sex of the embryos was inferred based on whether they exhibited a territory (females, *n*=17) or not (males, *n*=8). (F) RNA-seq as in (B) Bar plots showing the mean RNA levels (*n*=3) of MSL orthologue genes along embryogenesis in males and females. The overlaid dots represent the individual Log10(TPM+1) expression values of a given biological replicate.

## Alterations of H4K16ac on the Z chromosome of old females

While conducting H4K16ac immunofluorescence stainings in epithelial cells of the abdomen, we observed variations among adult female individuals of different ages (Fig 5A). In 25-days-old females, the majority of individuals displayed

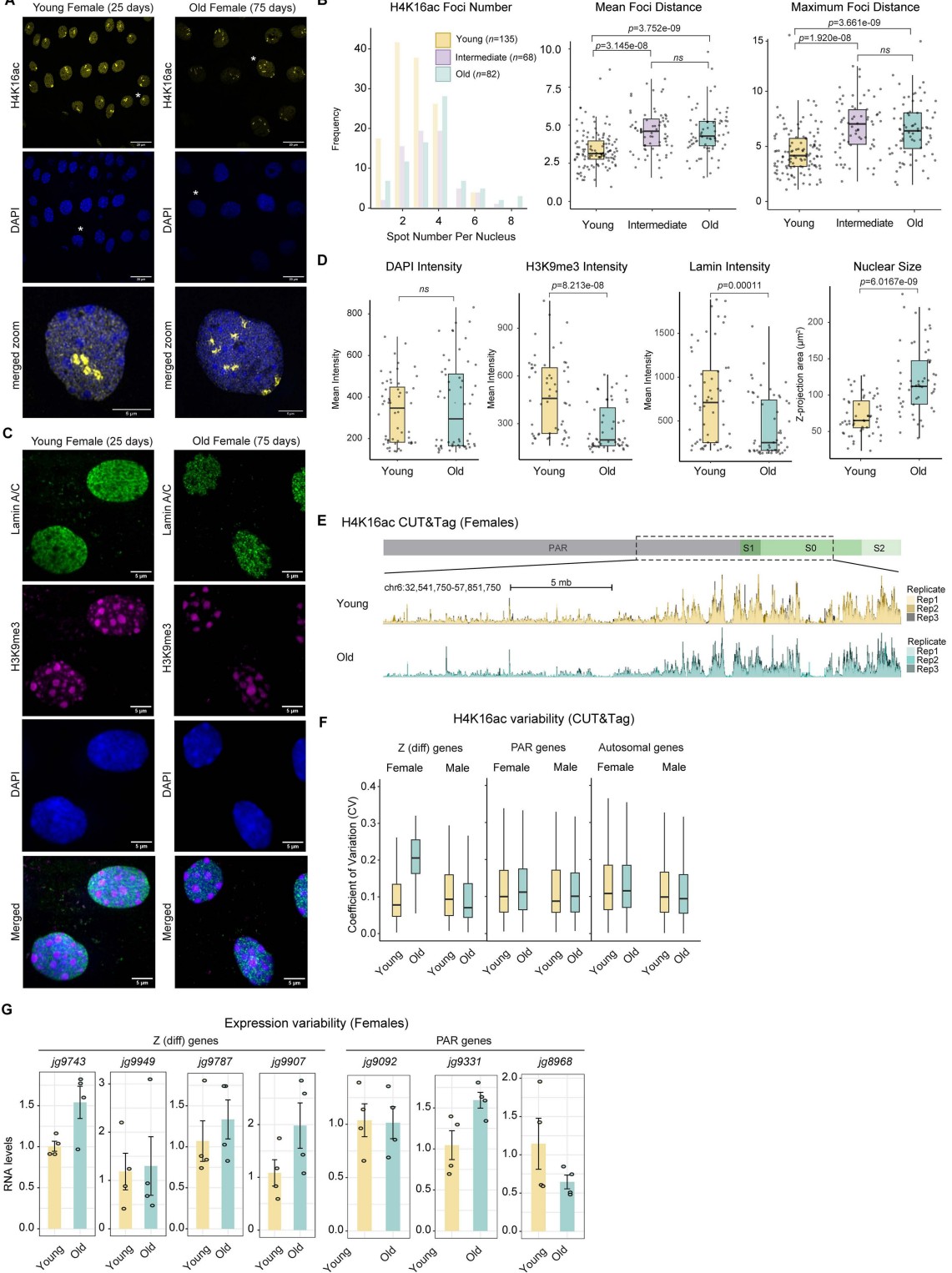

**Fig 5. Changes in nuclear organization and H4K16ac upon aging.** (A) Immunofluorescence staining of H4K16ac (yellow) and DAPI (blue) in young (25 days), and old (75 days) females. Cells marked with an asterisk are shown in the magnified images in the bottom row. For 25-day-old individuals, $n = 3$ individuals were analyzed, each with three images per individual used for foci counting and measurement. Two individuals at an intermediate

age (55 days) were analyzed, with three images quantified per individual. Similarly, for 75-day-old individuals ($n = 2$), three images were quantified per individual. (B) (left:) Histogram showing the frequency distribution of H4K16ac foci numbers per cell from young (135 quantified cells), intermediate (68 quantified cells) and old (82 quantified cells) females. The $p$-values obtained with a generalized linear model for Poisson-distributed data for comparisons between young and intermediate individuals ($p = 0.015839$) and between young and old individuals ($p = 0.000246$) were statistically significant, whereas the comparison between intermediate and old individuals was not significant ($p = 0.350181$). (right:) Box plots showing mean and maximum distances between foci within a cell, with each dot reflecting the measurement in one cell. $P$-values: Wilcoxon rank-sum test (C) Immunofluorescence staining of Lamin A/C (green), H3K9me3 (purple), and DAPI (blue) in young (25-day-old) and old (75-day-old) females. $N = 2$ individuals per age group, with three images analyzed per individual. (D) Boxplots showing the mean intensity levels of Lamin A/C, H3K9me3, and DAPI in the nucleus as well as nuclear area, in cells from young ($n = 136$) and old ($n = 95$) individuals. Z-projections were used for quantifying both intensity levels and nuclear area. (E) H4K16ac CUT&Tag data comparing young and old individuals. The genome browser snapshot displays a region from PAR and Z(diff), with the 3 replicates shown as overlays with different shades. (F) CUT&Tag as in (E) showing a boxplot of the variability of H4K16ac levels on expressed genes (TPM ≥ 10) as the coefficient of variation (CV, Standard Deviation/ Mean of each gene across individuals). The H4K16ac enrichment levels in each replicate and sex were calculated using multiBigWigSummary from the normalized signals in CPM spanning the gene body length as well as 1 kb upstream of the TSS. $p$-values were calculated using a two-sided Wilcoxon rank-sum test with Bonferroni correction for multiple comparisons. Autosomes ($n = 5956$ genes): young vs. old females, $p = 0.0015$; young vs. old males, $p = 0.0319$. Z(diff) ($n = 102$ genes): young vs. old females, $p < 2e-16$; young vs. old males, $p = 1$. PAR ($n = 330$ genes): young vs. old females, $p = 1$; young vs. old males, $p = 1$. (G) Bar plots showing the mean RNA levels of the indicated genes in young and old females ($n = 4$ biological replicates each, indicated by the overlaid data points) as quantified by RT-qPCR. The geometric mean of two reference genes (jg15373, jg3163) was used for normalization and the data expressed relative to the young individuals.

1 or 2 foci per nucleus (Fig 5B). We suppose that these numbers reflect different cell cycle phases (G1 = 1 dot, G2 = 2 dots). By contrast, mid-age (around 55 days) and old-age individuals (around 75 days of age; mean lifespan of females is around 45 days; see below) displayed on average 4 foci per nucleus with some nuclei showing up to 8 foci. We also noticed that when multiple foci were present in a given nucleus, there was a declustering indicated by a significant increase in the mean and maximum distance between H4K16ac foci in mid-age and old females (Fig 5B, right panels).

To investigate whether this is a general aging effect or specific to H4K16ac, we analyzed heterochromatin changes by Histone 3 lysine 9 trimethylation (H3K9me3) staining, as well as overall nuclear integrity by Lamin A/C staining (Fig 5C). In line with heterochromatin loss being a hallmark of aging [62], the H3K9me3 intensity decreased significantly (Fig 5D). In addition, the Lamin intensity decreased (Fig 5D) with the staining becoming more granular (Fig 5C) and there was an increase in nuclear volume.

To quantify the H4K16ac changes genome-wide, we conducted CUT&Tag in three young and three old individuals of both sexes (mapping rates and quality control in S1 Data). This data showed that the average levels of H4K16ac on genes on the Z(diff) region and autosomes remain similar when individuals age (Figs 5E and S6A). Genes in the PAR region showed no changes in males, but a modest decrease in old compared to young females (S6A Fig). This indicates that the additional foci observed by microscopy do not correspond to larger genomic areas of *de novo* acetylation, but - in line with the changes in Lamin and heterochromatin - reflect a general deterioration in the spatial organization of chromosomes in old individuals.

Interestingly, between old individuals the H4K16ac levels on the female Z(diff) became more variable, as indicated by a significant increase ($p < 2e-16$) in the coefficient of variation (CV, standard deviation divided by the mean CUT&Tag signal per gene across individuals) (Fig 5F), which was independently confirmed with a principal component analysis (S6B Fig). Analysis of the CV showed that this effect was specific to the female Z(diff): neither female autosomes nor Z-PAR regions, nor any region in males, exhibited a substantial change in H4K16ac variability between young and old individuals. In addition to H4K16ac, we detected age-associated changes in gene expression variability in several Z(diff) and PAR genes (Fig 5G).

Since epigenetic alterations are considered a primary hallmark of aging, and according to that classification [62], progressively accumulate with time and contribute to the aging process, we next wondered whether there are differences in male and female lifespan. Indeed, the presence of heteromorphic sex chromosomes has been associated with influencing lifespan (i.e., males in XY species such as *Drosophila* tend to be more short lived, while females in ZW species such as *Crotalus* (rattlesnake; are more short lived [31, 63–65]). Measuring the lifespan of *Artemia*, we found that ZW females

are indeed more short-lived than males, independently of whether the male and female individuals were reared together (i.e., allowed to mate), or, separately (Figs 6A and S7A). Since histone acetyl-writer and eraser factors have received considerable attention in the context of lifespan modulation [66–69], we assessed the expression levels of these factors in young and old individuals (Fig 6B). This revealed that several of them change expression but also show increased variability in old females, providing one possible explanation of the perturbed histone acetylation homeostasis (Fig 5F). Of note, we detected a significant increase in expression of the Sirtuin-type histone deacetylases *SIRT1* (*jg8800*) in old females (Fig 6B). Comparison of the 3D structures of human and *Artemia* SIRT1 catalytic domains using AlphaFold and FoldSeek, as well as regular sequence alignments, indicated that the proteins remain highly conserved and thus, likely function similarly (S7B-S7D Fig).

Lastly, we sought to determine whether the variability in acetylation and gene expression is a general principle observed and shifted our focus to *Drosophila*. While there are no sex-specific H4K16ac profiles in aging flies, there is an informative single-cell RNA-seq dataset [70]. Among many other findings, this study found that global transcriptional output is reduced in old individuals [70, 71]. Thanks to the UMI counts, this absolute expression decline per cell can be quantified by a negative Pearson correlation of the mean expression level with age [70] (plotted for males and females

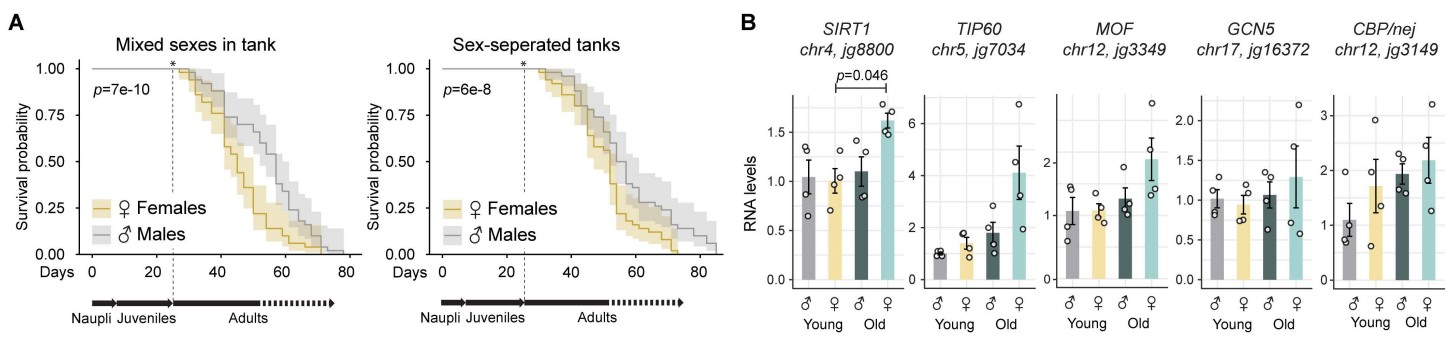

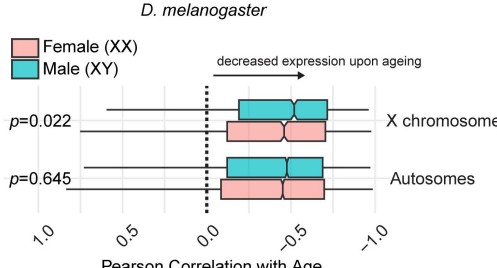

**Fig 6. Sex-specific lifespan and gene expression.** (A) Line plot showing the mean survival probability with shaded confidence intervals of male and female *Artemia* when they are reared (left:) in a mixed-sex culture or (right:) in separate cultures (no mating). Each replicate culture reflects 50 adults of each genotype seeded. The *x*-axis corresponds to the days after hatching. The experiment was started by culturing the same number of adult stage males and females (indicated by an asterisk), as before the sexes cannot be visually distinguished. The two other independent experiments are shown in S7A Fig. The *p*-value was obtained with a log-rank test for stratified data (Mantel-Haenszel test). (B) Bar plots showing the mean RNA levels of the indicated genes in young and old females (*n* = 4 biological replicates each, indicated by the overlaid data points) as quantified by RT-qPCR. *jg15373* was used for normalization and the data expressed relative to the male young individuals. *p*-values were calculated for each gene with a pairwise *t*-test (two-sided) with Holm–Bonferroni correction for doing multiple comparisons. Except for *SIRT1* (old vs. young females) no *p*-values <0.05 were obtained. (C) Boxplots showing the global decline in RNA levels across aging in *D. melanogaster* for the different chromosomal arms and sexes. The decline was quantified as the Pearson correlation coefficient of the mean expression with age. A value of 0 would indicate no correlation (i.e., stable expression across aging), while a negative correlation is indicative of decreased mean RNA levels. Data from [70]. The *p*-value was obtained using a two-sided Wilcoxon rank-sum test without continuity correction, with Bonferroni correction applied to adjust for multiple comparisons.

separately in S6C Fig). MSL and other histone acetylation factors were not unusual in that regard, as their decline did not differ significantly from other genes with similar expression levels (S6C-S6D Fig; statistical analysis in methods). However, when asking whether expression decline differs between males and females, and/or depends on chromosomal location, we interestingly found that X-linked genes of XY males show more negative Pearson correlation values compared to XX females (Fig 6C). Conversely, autosomal genes showed no differences between males and females. Altogether, our findings are consistent with the sex chromosomes in the heterogametic sex (ZW *Artemia* females, XY *Drosophila* males) showing changes that are distinct from the homogametic sex. We speculate that one factor contributing to these changes could be the perturbations in H4K16ac, which - besides broad effects on autosomes in both sexes - would affect the X-linked genes in *Drosophila* males and the Z-linked genes in *Artemia* females.

## Discussion

Our finding of H4K16ac-mediated DC in *Artemia* suggests that this histone modification was convergently recruited to regulate sex chromosome expression in different lineages characterized by different sex chromosome systems. Previously identified in XY lizards [24, 60] and XY *Drosophila* flies, our study now reports H4K16ac in a ZW crustacean species, thus extending earlier findings in the ZW monarch butterfly [13]. Despite the widespread presence of ZW sex chromosomes, their DC mechanisms remain largely unexplored. In chicken, the "model" ZW species, DC between the sexes is not achieved by a single, global mechanism, but rather a combination of gene-by-gene transcriptional regulation [9] along with post-transcriptional mechanisms [10, 11]. Our study is among the first to identify chromatin-linked modifications associated with this process in a ZW species. In monarch butterfly, H4K16ac is restricted to only the small neo-Z chromosome with unknown developmental dynamics and mechanistic basis. The ancient lepidopteran Z chromosome in turn appears to employ a different, yet-to-be-identified mechanism for balancing transcription. In *Artemia*, we find H4K16ac being highly enriched across the Z(diff), but not PAR region. This confinement raises questions about what prevents the spreading of this mark. Possibilities are the presence of boundary elements between Z(diff) and the PAR, or, entry elements akin the HAS (*D. melanogaster*) solely present on the Z(diff). A putative HAS-like motif that is AT-rich and previously associated with binding by the RNA-binding protein HuR, was recently reported to be enriched at *Artemia* H4K16ac peaks and within the Z(diff) DNA sequence [35]. It remains to be tested which factors bind that motif, whether the HuR-motif loci produce RNA, and whether/ how binding by the so far unknown upstream factors contribute to DC. Beyond a HAS-based mechanism, another possibility is the action of a non-coding RNA that recruits H4K16ac modifiers only to the Z(diff), perhaps by limited diffusion [72]. It is also possible that a combination of these molecular mechanisms may contribute to the observed pattern.

It is interesting to observe the conservation of a core part of the DC pathway (H4K16ac), but variability in its upstream regulators, which mirrors hourglass models well described for sex determination pathways [73]. In sex determination, different upstream regulatory factors ultimately funnel through sex-specific regulation of the *transformer* RNA. Similarly, DC mechanisms may also vary in their upstream regulators (acetylation writers, erasers and regulatory factors) while still converging on H4K16ac deposition on sex chromosomes. In *Drosophila*, the MSL-MOF complex is responsible for this histone modification, but in *Artemia* and other species (*Anolis*, monarch butterfly), different, as-yet-unknown factors may serve this role. We find that none of the MSL complex subunits show sex-specific expression in *Artemia*, nor are any of the components strongly induced during the onset of DC in embryos. In *Anolis*, the *cis*-acting non-coding RNA *MAYEX* does not interact with MOF, but rather a G1KD42 acetyltransferase [60] and a sex-specific expression of the TTRRAP-KAT5-APBB1 (TIP60) histone acetyltransferase regulatory complex has been found, rather than MOF-MSL [24]. While these factors remain to be functionally and mechanistically validated, these findings beautifully demonstrate, along with our work in *Artemia*, the complexity of species-specific adaptations in epigenetic regulation and suggest that multiple, independently evolved pathways can lead to similar chromatin modifications on sex chromosomes.

Beyond *Artemia*, our investigations across arthropods suggest that DC mechanisms independent of H4K16ac are certainly not uncommon. This is especially intriguing for the yet-to-be-discovered mechanism(s) that provide DC to insects with a deeply conserved X chromosome content which may reflect an ancient X that predates the origin of the class insecta [42]. This putatively "ancient" insect X chromosome is preserved in some beetles (including *L. decemlineata*) and stick insects (including *B. grandii*), but - as our data shows - does not rely on H4K16ac for compensation. This shows that H4K16ac is not the sole dominant mechanism across arthropods but rather one among multiple possible strategies for DC, thus being mechanistically highly flexible.

Lastly, we find a general deterioration of nuclear architecture and chromatin upon ageing in *Artemia*. This includes perturbed H4K16ac territories in females, accompanied by a shorter lifespan compared to males. In *Drosophila*, the opposite is seen with males being more short lived than females [74], which some studies attributed to Y chromosome toxicity [32, 33, 75]. However, a more recent study challenged this view, suggesting that male lifespan is shortened due to phenotypic sex rather than the presence of the Y chromosome [34]. Although the contributions of the MSL writer complex are still unclear, the H4K16ac eraser Sir2/SIRT1 [76] influences fruit fly lifespan differently in males and females [77] and we find SIRT1 deregulation in old female *Artemia*. In *C. elegans*, H4K16ac is implicated in both DC (reduced on the hermaphrodite X) and lifespan regulation [78]. Together, this points towards the possibility that H4K16ac and/or DC may influence sex-specific aging processes. Mechanistically, this could be linked to mechanisms like epigenetic noise and transcriptional instability [79], observed by us here in *Artemia* but also found in other systems like mammals (e.g., in the immune system [80]). In mammals, loss of DC upon aging has been linked to both beneficial and deleterious outcomes. Age-related escape from X inactivation can lead to biallelic expression in XX females - versus single-copy expression in XY males - and has been associated with improved cognitive function and protection against pulmonary fibrosis [81–84]. Conversely, the age-associated escape of *Tlr8* could promote autoimmune conditions like late-onset lupus [85]. This raises the possibility that while DC is critical early in life for correcting dosage imbalances, its modulation later in life could be beneficial, but could also create an "epigenetic vulnerability" affecting sex-specific traits and disease risk.

Taken together, we reveal an unanticipated complexity in H4K16ac and DC across species, its role in different contexts perhaps explaining why this mechanism is not a universally preferred solution. It will be interesting to identify the upstream regulators of H4K16ac in *Artemia* and novel mechanisms acting in other species, investigating the presence of *cis*-acting non-coding RNAs, and exploring how H4K16ac influences sex-specific phenotypes across diverse taxa.

## Materials and methods

### Arthropods

*Drosophila melanogaster* were raised on a standard molasses-based food at 25°C, 55% humidity and 12:12 hours light-dark cycle. Male and Female flies, 2–3 days after eclosion, were used for the experiments.

*Anopheles gambiae* mosquitoes were maintained in standard insectary conditions at 26–28°C, 75–80% humidity and 12:12 hours light-dark cycle. PFA-fixed male and female tissues were kindly provided by Dr. Eric Marois (University of Strasbourg, France).

*Bacillus grandii stock cultures* were maintained on field-collected bramble cuttings (*Rubus* ssp.) at 25°C, 60% humidity and 12:12 hours light-dark cycle.

*Plodia interpuctella* Pi_Fog w-/- cultures were originally obtained from Arnaud Martin (George Washington University) and kindly provided to us by Dr. Ines Drinnenberg (Institut Curie, Paris). They were maintained in dedicated containers at 29°C, 70% humidity on a layer of coarse wheat bran, brewer's yeast flakes, dextrose, glycerin and canola oil.

*Reticulitermes flavipes* lab colonies, originally collected in France by Dino McMahon and Cedric Aumont (FU Berlin), were kept in climate chambers at 27°C, 80% humidity in constant darkness and fed on pine wood and filter paper.

*Leptinotarsa decemlineata* were maintained in a climate chamber with 24 °C, 70% humidity and a 16:8 hours light-dark cycle. Larvae, as well as adults, were fed with leaves from organically grown potato plants (Annabelle variety, purchased from Ellenberg's Kartoffelvielfalt GmbH & Co. KG, Barum, Germany), similar to [86].

*Bradysia* (formerly *Sciara*) *coprophila* flies of the Holo2 strain were kept at 18°C, 70% humidity in 2.2% agarose vials and fed with mushroom and spinach powder, baker's yeast, and oat straw. Before dissection, the flies were anesthetized with $CO_2$.

*Triops cancriformis* (Spanish population [45]) eggs were obtained from triopsking.de. Triops were reared at 23°C in deionized water and a 12:12 hours light-dark cycle. They were fed pre-prepared Triops food (triopsking.de) every three days until reaching adulthood. The independent evolution of the Z chromosomes of *Triops* and *Artemia* is likely, but not unequivocally proven, based on their phylogenetic distance and the data in [87].

(for A. franciscana, see next section)

### *Artemia* strains and rearing

*Artemia franciscana* cysts were obtained from the Institute of Aquaculture Torre de la Sal (IATS, Spain). Nauplii were hatched from cysts in water supplemented with synthetic sea salt (Aquatistik Shop, 552001) at a concentration of 30 g/L under constant aeration. Adult *A. franciscana* were maintained in water supplemented with synthetic sea salt at a concentration of 60 g/L. All *A. franciscana* individuals were kept at 28°C with a 12-hour light/dark cycle and fed with *Tetraselmis suecia* (Phytobloom) every two to three days. To obtain embryo samples at specific developmental stages (for RNA-seq, immunofluorescence), single unfertilized females were paired with two males and monitored every two hours for fertilization using a dissecting microscope to determine the time of fertilization which is indicated by fusion of the lateral brood pouches as described in [88]. Following fertilization, females were separated from the males and dissected at the desired time-points. To ensure that the embryos developed ovoviviparously, the shell glands were examined under a dissecting microscope. A brownish coloration was used as an indicator of development via the oviparous pathway (cysts).

### Lifespan assays

*Artemia* cysts were hatched and cultured for 25 days, allowing them to reach adulthood. They were then distributed into 4-liter tanks with an initial population of 50 individuals per tank. Two tanks contained a mixed-sex population (25 males, 25 females each), while either 50 males or 50 females were reared in separate tanks. Each condition was tested in triplicates, and data obtained from the two mixed-sex tanks per replicate were combined for appropriate comparison with the single-sex tanks ($n = 50$ initial number of individuals). As the number of surviving individuals decreased, the water volume in each tank was reduced proportionally to maintain consistent density and minimize effects of population density on lifespan. Mortality was monitored every two to three days by counting the remaining individuals in each tank. Offspring produced in the mixed-sex tanks were removed at the nauplius stage to prevent confounding effects on the experiment. The *Artemia* were fed every second day with *Tetraselmis suecia* freeze dried algae, diluted in salt water, ensuring no nutrient restriction. The data was analyzed and visualized using the survminer package (https://github.com/kassambara/survminer) in R. Statistical difference was evaluated with a stratified Mantel-Haenszel test.

### CUT&Tag

Individual *Artemia* were transferred to ice-cold PBS to remove excess salt, then dissected to remove gonads and guts using forceps. Excess liquid was removed, and samples from single individuals were stored at -80°C until use, or directly processed (see legends). A cell suspension was generated using a handheld pestle in 60 µL of PBS, where half of the volume was used for CUT&Tag library preparation. Cells were pelleted by centrifugation (2000 x g) and resuspended in 100 µl wash buffer (20 mM HEPES pH 7.5, 150 mM NaCl, 0.5 mM spermidine, 1x protease inhibitor (Sigma P8340) in nuclease-free $H_2O$). For *D. melanogaster* 5 heads were dissected and they were homogenized using a handheld pestle

directly in 100 µl wash buffer and this suspension was passed through a cell strainer. For *B. grandii*, the head was cut off with a scissor and homogenized in 100 µl PBS, before passing through a cell strainer (2000 x g) and resuspending the pellet in 100 µl wash buffer. Concanavalin A-coated magnetic beads (Polysciences, 86057) were washed twice in binding buffer (20 mM HEPES pH 7.9, 10 mM KCl, 1 mM $CaCl_2$, 1 mM $MnCl_2$) before beads were added to the cells and incubated rotating for 10 minutes at room temperature. The binding buffer was removed and the bead-cell-samples was resuspended in 50 µL of antibody buffer (wash buffer containing 0.05% Digitonin (Millipore, 300410), 0.1% BSA (NEB, B9200S), 2 mM EDTA). Primary antibodies, H4K16ac (Millipore, 07–329), RNA polymerase II (Active Motif, 39097) and IgG control (abcam, AB37415), were prepared in antibody buffer at a dilution of 1:100. The antibodies were added to each bead-cell aliquot and incubated overnight at 4°C on a nutator. After antibody incubation, samples were rinsed with dig-wash buffer (wash buffer containing 0.05% Digitonin). The secondary anti-rabbit antibody (antibodies-online, ABIN101961) was prepared 1:100 in dig-wash buffer. 50 µL of antibody solution were added to the bead-cell aliquots and placed on a nutator for 1 hour at room temperature. Following the incubation, the liquid was removed from the beads and they were rinsed and washed once with dig-wash-buffer. The mosaic end oligo duplex loaded Tn5-transposase enzyme (produced by the IMB Protein Production Core Facility) was mixed with dig-300 buffer (20 mM HEPES, 300 mM NaCl, 0.5 mM spermidine, protease inhibitor, 0.05% Digitonin) at a concentration of 1:100. 100 µL were added to each sample, followed by a 1-hour incubation at room temperature on a nutator. The samples were rinsed and washed once in dig-300-buffer after the incubation. Tagmentation was performed for 1 hour at 37°C in tagmentation buffer (dig-300 buffer, 10 mM $MgCl_2$). The reaction was halted by adding 1.5 µL of 10% SDS, 5 µL of 0.5 M EDTA, and 1.25 µL of 20 mg/mL proteinase K (Life Technologies, AM2546) to each sample, followed by a 1-hour incubation at 55°C. DNA was purified using the DNA Clean & Concentrator-5 kit (Zymo Research, D4014). Libraries were amplified using the NEBNext Ultra II Q5 Master Mix (New England BioLabs, M0544) with 15 cycles. After cleanup with SPRI beads, the fragment size and integrity were assessed using a High Sensitivity DNA chip on a 2100 Bioanalyzer (Agilent technologies). The libraries were quantified with the Qubit dsDNA HS Assay kit in a Qubit 2.0 Fluorometer (Life Technologies) and pooled in equimolar ratio. Paired-end sequencing was performed on a NextSeq 2000 by the IMB Genomics Core Facility (SR Read with 2x111bp + 2x8nt for the dual index read).

### RNA isolation and RT-qPCR

Adult *Artemia* individuals were dissected with forceps and the germline was removed. The dissected samples were put in 50 µL of RNA shield (Zymo Research, R1100) and stored in a -80°C freezer until use. For the embryo RNA-seq, individual embryos staged as described in the section "Rearing" were transferred to Eppendorf tubes containing 50 µL of RNA Shield and stored at -80°C until further processing. RNA was extracted using the Direct-zol RNA MicroPrep Kit (Zymo Research, R2062) with 150 µL TRIzol (Fisher Scientific, 15-596-026). Quantification was carried out using the Qbit assay. For qPCR quantification, cDNA was synthesized with olido(dT) primers with a reverse transcriptase enzyme produced by the IMB Protein Production Core Facility. qPCR was done in a 7 µL reaction at 300 nM final primer concentration with the FastStart Universal SYBR Green Master (ROX) mix (Roche, 04913850001). The reaction was run in the LightCycler 480 2 (Roche, 05015243001) with the cycling conditions as recommended by the manufacturer. Serial dilutions were used to test primers and all showed results in the range of 90–105% for their efficiency without amplification in water control.

### Primer sequences

| ID | Dir | Sequence | Gene | Comment |
|---|---|---|---|---|
| q876 | fwd | AAGGAAGGAGCTTGACAACG | jg15373 | reference gene |
| q877 | rev | AGCCAGTAAAGCAAAATCAGCA | jg15373 | reference gene |
| q878 | fwd | CTACTGAAAGTCCCGTGCCA | jg3163 | reference gene |
| q878 | rev | TGACGGAACAACAGGGGATG | jg3163 | reference gene |
| q984 | fwd | CCGAAACTCTGGGTGTGTGA | jg3349 | *MOF* |

| ID | Dir | Sequence | Gene | Comment |
|---|---|---|---|---|
| q985 | rev | ACACCAAGTGCATTGACCCA | *jg3349* | *MOF* |
| q992 | fwd | AGTTCAAGCCTTCGCTCTGT | *jg8800* | *SIRT1* |
| q993 | rev | GCGCTGAATCTCTGCTACCT | *jg8800* | *SIRT1* |
| q1002 | fwd | CCAAAAGAACGCAACCACCA | *jg7034* | *TIP60* |
| q1003 | rev | CGGCTTGAAGAACGTTTGGG | *jg7034* | *TIP60* |
| q1006 | fwd | TGTCAAACACGGCATGCTTC | *jg16372* | *GCN5* |
| q1007 | rev | TGTAGCCACCATACACGGTT | *jg16372* | *GCN5* |
| q998 | fwd | TCAAGGCCCAAGAATGCAGA | *jg3149* | *CBP/nej* |
| q999 | rev | TGGTTTGACAGACCTCCAGC | *jg3149* | *CBP/nej* |
| q1018 | fwd | TCGACGGGAAGATCAGTCAG | *jg9743* | |
| q1019 | rev | ATAAGCGCGTCGGTTAGAGG | *jg9743* | |
| q1020 | fwd | TGGGAAGATACCAGAGCACG | *jg9787* | |
| q1021 | rev | ACATGGCCTGCTTCAACTGT | *jg9787* | |
| q1028 | fwd | GCTGCAGTCGTTTAGTACTCG | *jg9907* | |
| q1029 | rev | TTGCGTAGAGAGCAACATGC | *jg9907* | |
| q1032 | fwd | CGGCGGTTCTTCAACCTTTC | *jg9949* | |
| q1033 | rev | CCCCATGCCCTTTCAGAAGA | *jg9949* | |
| q884 | fwd | TGTGTTGCACCGTTCCATTTTG | *jg8968* | |
| q885 | rev | TACTCAGTGGTGGCAGGAACT | *jg8968* | |
| q886 | fwd | TGGTTTTTCTGACAGATCCGA | *jg9331* | |
| q887 | rev | TTGGGCTGCAAATCTTGTGG | *jg9331* | |
| q890 | fwd | GCGTCCTTTGACCCGAAAAG | *jg9092* | |
| q891 | rev | TGAGCGAGGAATTGACATCAT | jg9092 | |

## Genome assembly and regions

As a reference genome, the assembly available at [47, 89] was used. Genes were categorized into autosomal genes, pseudoautosomal (PAR) genes on the Z, and genes from the differentiated region of the Z chromosome, referred to as Z(diff). The genomic regions of the Z chromosome were classified based on [47] specifically: (46,085,001–48,385,001 bp) (S1), (48,665,001–53,365,001 bp) (S0), (53,585,001–54,575,001 bp) (S0), (54,715,001–61,855,001 bp) (S0), and (63,345,001–64,075,001 bp) (S2). All other regions of the Z chromosome, outside these specified ranges, were categorized as pseudoautosomal regions (PAR).

## poly(A) RNA-seq library generation

Libraries were prepared using Illumina's Stranded mRNA Prep Ligation Kit following the manufacturer's protocol. A starting amount of 65 ng was used, with 15 PCR cycles for amplification. Two post-PCR purification steps removed residual primers and adapter dimers. Libraries were assessed using a High Sensitivity DNA chip on an Agilent 2100 Bioanalyzer and quantified with the Qubit 1x dsDNA HS Assay Kit on a Qubit 4.0 Fluorometer. All 16 samples were pooled in equimolar amounts and sequenced on a NextSeq2000 P3 (100 cycles) FC, SR for 1x 116 cycles plus 2x 10 cycles for the dual index read and 1 dark cycle upfront R1.

## RNA-seq data analysis

Publicly available transcriptome sequencing datasets of pooled male and female *Artemia franciscana* adults (SRR8641211, SRR8641212, SRR8641215, SRR8641216) [14] were downloaded as FASTQ files from the Sequence Read Archive (SRA). Each dataset consisted of pooled RNA from the heads of five adult individuals. Reads were processed using TrimGalore

(v0.6.10) for trimming and were mapped to the reference genome using STAR (v2.7.3a) with default settings. Transcript quantification was performed with Salmon quant (v1.9.0), using the --gcBias and --validateMappings options to account for GC bias and ensure accurate read mappings. Genes were classified as "expressed" if they had a TPM value of ≥10 in either the male or female dataset and "non-expressed" if the TPM was <10 in both sexes.

For the *Drosophila* single-cell RNA-seq from adult fruit flies of different ages (8 time-points up to 50 days) [70] the dataset 'Aerts_Fly_AdultBrain_Filtered_57k.loom' containing count data for 17.473 genes from 56.902 cells was downloaded from https://scope.aertslab.org. Cells were grouped by sex and age according to the provided meta-data. For each group, counts were summarized per gene to determine mean count, standard deviation, coefficient of variation and fraction of cells with non-zero counts. Per gene decline with age was assessed by calculating the Pearson correlation of gene expression with age in R. To analyze whether MSL orthologues and histone acetylation factors show a different degree of decline than an average gene with similar expression level, we performed a permutation-based outlier detection. For this, we first generated a null distribution of Pearson correlation values by sampling 100 random genes with similar mean expression levels (± 10%) and ranking their Pearson correlations. Then, we determined if the Pearson correlation of our gene-of-interest is significantly lower than expected by calculating its percentile rank. The *p*-value was obtained as a proportion of sampled correlations being more negative (or more positive) than the gene-of-interest. The result was that none of the tested genes-of-interest, except for the positive control *sta* [70], showed a *p*-value <0.05.

To investigate expression of MSL orthologues across species, we used the raw read counts provided as Supplementary File in [90] for *A. gambiae*, and the raw read counts from [57] for *D. melanogaster*. We then ran DESeq2 to obtain normalized read counts, which were plotted in the figures. For *L. decemlineata*, we obtained already normalized expression values from [86].

## Developmental RNA-seq library generation

cDNA generation and NGS library prep was performed with Takara's SMART-Seq mRNA LP (with UMI) Kit (100422) following the manufacturer's recommendations. The adult samples were normalized to 0.05 ng of total RNA for cDNA generation. All other samples had an input volume of 9.5 µL with unknown concentration (below detection limit) of total RNA. cDNA was amplified by 18 cycles of LD-PCR. Libraries were prepared with a starting amount between 0.1–1 ng of cDNA and were amplified in 16 PCR cycles. Libraries were analyzed with a High Sensitivity DNA Chip on a 2100 Bioanalyzer (Agilent technologies) and quantified using the Qubit 1x dsDNA HS Assay Kit, in a Qubit 4.0 Fluorometer (Invitrogen by Thermo Fisher Scientific). Samples were pooled in equimolar amounts and then purified two times with AMPure XP Beads (Ratio 0.8X) and sequenced on a NextSeq2000 P3 (200 cycles) FC, PE for 2x 111 cycles plus 2x 8 cycles for the dual index read.

## Developmental RNA-seq analysis

Sequencing data was analyzed using the Cogent NGS Analysis Pipeline v2.0.1 provided by Takara Bio Inc on its website (https://www.takarabio.com) as recommended by the manufacturer. The tool is proprietary by Takara Bio Inc but the command line code to run the tool can be provided upon request. Non-demultiplexed FASTQ files were generated with bcl-convert v.4.1.5 from Illumina as requested for compatibility with the Takara pipeline. Subsequently they were demultiplexed per sample using the 'cogent demux' call of the pipeline. A reference genome was generated with 'cogent add_genome' based on the genome assembly provided on [89]. Read trimming, genome mapping and gene expression counting was applied as implemented in the pipeline's 'cogent analyze' functionality (experiment type set to 'SMARTSeq_FLA_UMI'). The resulting gene counts obtained from all reads were used for differential expression analysis with DESeq2.

## Intron Retention analysis of early embryo RNA-seq data

Aligned bam files of 12 embryo samples at 1–2 dpf (3 male/ 3 female) and 4 dpf (3 male/ 3 female) were assigned to detection of intron retention (IR) with the IRFinder tool (v.1.3.1) [61]. The genome reference for IRFinder was built using IRFinder 'BuildRefProcess' command on the *Artemia franciscana* reference genome provided by [47]. The provided

annotation gtf file had to be slightly modified to match IRFinder's requirements (use attribution tags 'transcript_biotype' or 'gene_biotype' instead of 'original_biotype' and 'gtf' instead of 'gff' format). Intron retention events were quantified with IRFinder in 'BAM' mode. Having >=3 replicates per condition, we used the Generalized Linear Model (GLM) approach with R-package DESeq2 (v. 1.36.0) [91] for differential IR analysis as described in the IRFinder manual. Differential analysis was done for female vs. male samples in each of the 1–2 dpf and 4 dpf sub-groups ($n$ = 3) and as well in the combined 1–4 dpf group ($n$ = 6).

## Bin-wise differential expression analysis to explore unknown transcripts

We applied an approach described by Tenorio et al. [60] to explore genomic regions with potentially differentially expressed transcripts not included in current genome annotations. The *Artemia* franciscana reference genome provided by [47] was divided into bins of 50 bp using bedtools (v.2.29.2) [92] 'makewindows' command. Bedtools 'coverage' command was used to compute the coverage per bin for each aligned bam file. The resulting counts per bin data was used as a count matrix for subsequent differential expression analysis between male and female samples ($n$ = 6) using the R-package DESeq2 (v.1.36.0) [91]. The setting 'independentFiltering=TRUE' was set to test for informative intervals only. Differential expression $p$-values adjusted for multiple testing were plotted as Manhattan plot against the genomic location to identify regions with significant signal not overlapping the previously applied genome annotation.

## Sex determination from expressed SNPs/ Haplotype calling

The mechanisms of sex determination and establishment of sex-specific gene expression in *Artemia* are entirely unknown, neither are there any marker genes for sex determination. In addition, early embryos show no visual differences between sexes. Therefore, we determined the sex from expressed single-nucleotide polymorphisms (SNPs) assuming that females (ZW) should exhibit fully monoallelic genes in the Z(diff) region compared to the ZZ males with two copies of Z(diff) genes. For this, we used the mapped bam-file for haplotype calling on chr6 (Z chromosome) using freeBayes [93]. The following options were used: freebayes --fasta-reference "${REFERENCE_GENOME}" \ --region "${REGION}" \ -b "${BAM_FILE}" \ --vcf "${OUTPUT_DIR}/${BASE_NAME}_chr6.vcf" \ --min-coverage 5 \ --skip-coverage 0 \ --limit-coverage 0 \ -i -X -u \ -n 0 \ --haplotype-length 3 \ --min-repeat-size 5 \--min-repeat-entropy 1 \ --min-mapping-quality 1 \ --min-base-quality 0 \ --min-supporting-allele-qsum 0 \ --min-supporting-mapping-qsum 0 \ --read-indel-limit 1000 \--min-alternate-fraction 0.1 \ --min-alternate-count 1 \--min-alternate-total 1. We then used bedtools intersect [92] to only keep the SNPs overlapping with exons (a subset of obtained results after this step for each sample are plotted in S2 Data). After filtering for mapping quality (MQMR > 20, MQM > 20) and read coverage (AO > 10, RO > 10), we allocated the SNPs before chr6 region <46043529 to PAR and the ones with >46043529 to Z(diff) region (S5D Fig). Note that SNPs were also returned due to differences in our population compared to the reference genome and because our population is non-isogenic. Therefore, we compared the distribution of allelic balance (AB) of each SNP on the Z(diff) in comparison to the PAR region, which is still freely recombining. As positive control, we used adult samples with known sex, where ZZ males did not show differences in allelic balance for Z(diff) in comparison with PAR (S5E Fig). We also investigated the SNP count, based on the condition AB > 0 ("biallelic"), else "monoallelic" (NA were removed). ZW females (which exhibit monoallelic expression of Z(diff) - linked genes, when the W is degenerated) show a lower number of biallelic and higher number of monoallelic expressed genes in the Z(diff) (S5F Fig, bottom) contrasting ZZ males (S5F Fig, top).

## CUT&Tag data analysis

Paired-end reads were trimmed using TrimGalore (v0.6.10), and quality control was performed using FastQC. Reads were aligned to the reference genome using Bowtie2, and sorted and indexed with samtools (v1.10). Read duplicates were not removed as this is not recommended [94], because fragments that share exact starting and ending positions are expected to be common in CUT&Tag and likely arise from Tn5 enzyme characteristics, rather than PCR. BigWig files were generated

using deepTools (v3.5.5) bamCoverage with CPM normalization (--normalizeUsing CPM). For the metaplots a matrix was generated using deepTools (v3.5.5) computeMatrix function with the flags --regionBodyLength 5000 -b 1000 -a 1000 --sortUsing median. Metaplots were generated using deepTools (v3.5.5) plotHeatmap function using the flag --averageTypeSummaryPlot median. Log2FC changes between male and female H4K16ac levels were assessed using deepTools (v3.5.5) bigwigCompare command. BigWig files were combined using the deepTools multiBigWigSummary command with default of 10kb consecutive genomic bins for downstream analysis using R (v.4.2.2). For the PCA of H4K16ac levels on Z-diff genes, the H4K16ac signal was extracted from summary files generated with multiBigwigSummary (deepTools v3.5.5). Principal components were calculated using the prcomp function in R (v4.2.2). The first two components were visualized with ggplot2. Genome browser snapshots were generated with Integrative Genomics Viewer (IGV 2.17.4).

## CUT&Tag data analysis of *B. grandii*, *D. melanogaster* and public *A. franciscana* data sets

The bioproject PRJNA1150095 was downloaded from SRA using the SRA toolkit (v.2.11.0). Overall sequence quality and rRNA content of fastq files from all three data sets has been assessed with FastQC (v.0.11.9) (http://www.bioinformatics.babraham.ac.uk/projects/fastqc) and FastQScreen (v.0.15.2) [95]. Paired end reads were trimmed for Illumina adapter sequences using cutadapt (v.4.0) [96] and subsequently mapped with bowtie2 (v.2.4.5) [97] to the respective reference genome applying the '--very-sensitive' preset. For *D. melanogaster* we used NCBI genome assembly release 6 (dm6, GCA_000001215.4) as reference genome yielding a unique mapping rate >80%. In absence of an appropriate reference genome of *B. grandii* we used NCBI genome assembly Brsri_v3 of *Bacillus rossius redtenbacheri* (GCF_032445375.1) instead, yielding a unique mapping rate of ~25%. For *A. franciscana* we used the reference genome assembly provided by [89] obtaining ~38% unique mapping rate. Coverage tracks with CPM normalization were generated using the bamCoverage tool from deepTools software suite (v.3.5.1) [98]. BigWig summaries and average bigWig files per group using a 5kb bin size were generated with deepTools multiBigwigSummary and bigwigAverage, respectively. DeepTools bigwigCompare was used to generate log2 ratio tracks for female vs. male samples.

## Orthologue searches and alignments

Protein sequences of *A. franciscana* were obtained from [47]. Functional annotation and conserved protein domain identification were performed using the eggNOG-mapper web server (http://eggnog-mapper.embl.de/, accessed on 15 May 2024) with eggNOG version 5 as the search database. The following parameters were used: minimum hit e-value of 0.001, minimum hit bit-score of 60, percentage identity of 40%, minimum query coverage of 20%, and minimum subject coverage of 20%. Orthologs of MSL complex members in *A. franciscana* were identified using BLASTp searches based on protein domains from MSL complex members in *D. melanogaster*. Protein sequence alignments were generated with clustalo (1.2.4) using the web interface at https://www.ebi.ac.uk/ and aligned with ESPript [99]. The 3D structural alignment of the SIRT1 catalytic domain was generated on the foldseek web interface https://search.foldseek.com/search [100]. 3D alignments were generated and visualized with foldmason on https://search.foldseek.com/foldmason [51].

## GoTerm analyses

For GOterm analysis, we combined the information of two different gene annotations in order to increase the chance of retrieving meaningful GO results. We first performed functional annotation of proteins identified from the genome assembly from [47] and assigned GOTerms using the eggNOG-mapper web server with eggNOG version 5 as the search database (see previous section). In parallel, we ran the same pipeline on the proteins annotated in the NCBI RefSeq assembly GCF_032884065.1. The results from the two GO annotations were combined. For enrichment analyses, we used all genes with an assigned GO term as the background. The enrichment analysis of GOterms in DE genes was then performed with the enricher function in the clusterProfiler [101] R package (*p*-value cutoff = 0.05, pAdjustMethod = "BH").

## Immunofluorescence stainings

*B. coprophila* malphigian tubule were dissected and placed immediately into ice-cold phosphate-buffered saline (PBS). The tissue was fixed in 2% paraformaldehyde (PFA) in PBS for 15 minutes at room temperature and washed three times with PBX (phosphate-buffered saline with 0.3% Triton X-100). Adult *A. franciscana* were transferred to PBS and the gut was carefully removed to eliminate algae. To stain the abdomen, the thorax was removed, and the abdomen was inverted, followed by fixation. Incisions were made in the head using needles to expose internal tissues, followed by fixation. All other insect species (Fig 1) were dissected in PBS at adult stages followed by fixation as described in the following. Fixation was performed in 4% methanol-free paraformaldehyde (PFA) (Life Technologies, 28908) for 15 minutes, followed by washes in PBS with 0.2% Tween-20 (Sigma, P1379). Blocking was done for 1 hour with 0.5% bovine serum albumin (BSA) (Pan-Biotech, P06-1391500) and 0.3% Triton X-100 (Sigma, X100) in PBS. The samples were incubated overnight at 4°C on a nutator with primary antibody (H4K16ac (Millipore, 07–329), H3 (Active motif, 39064), Lamin A/C (Santa Cruz Biotechnology, sc-376248), H3K9me3 (Active motif, 39162) 1:400 in blocking buffer (BB; 0.5% BSA in PBS-0.2% Tween). After incubation, samples were washed four times with BB. The Alexa-fluorophore coupled secondary antibodies (ThermoFisher, A21430) were diluted 1:400 in BB. 200 µL were added to the samples and they were incubated for 2 hours at room temperature on a nutator. Samples were subsequently washed in PBS with 0.2% Tween-20, with a 10-minute incubation in 1:1000 DAPI (Life Technologies, D1306) solution. Samples were mounted in ProLong Gold antifade mounting media (Life Technologies, P10144).

Cryosections were prepared for embryo samples. For this complete brood pouches containing embryos were dissected in PBS and fixed in 4% PEM for 1 hour at room temperature. The pouches were then transferred to plastic cylinders and embedded in Tissue Plus Optimal Cutting Temperature (OCT) compound. Samples in OCT were frozen on dry ice and stored at -80°C until use. Cryosections of 12 µm thickness were prepared using a Leica CS3050 S cryostat. All subsequent steps were performed directly on microscopy slides within a humidity chamber. Permeabilization was carried out with 0.3% Triton X-100 (Sigma, X100) in PBS for 1 hour. The permeabilization buffer was then removed by washing with blocking buffer containing 0.2% Tween-20 (Sigma, P1379) and 0.5% BSA (Pan-Biotech, P06-1391500) in PBS. Samples were incubated overnight at 4°C with the primary antibody H4K16ac (Millipore, 07–329) and H3 (Active Motif, 39064) both at a 1:400 dilution in blocking buffer (BB; 0.5% BSA in PBS with 0.2% Tween-20). To prevent drying, samples were covered with small pieces of Parafilm. Following incubation, samples were washed four times with BB. Alexa Fluor-conjugated secondary antibodies (ThermoFisher, A21430) were diluted 1:400 in BB and incubated for 2 hours at room temperature. After incubation, samples were washed in PBS, followed by a 10-minute incubation in a 1:1000 dilution of DAPI (Life Technologies, D1306). Finally, samples were mounted using ProLong Gold Antifade Mounting Medium (Life Technologies, P10144).

## Microscopy

Images were acquired taking Z-planes using a BC43 Spinning Disk Confocal microscope (Andor) with 20X and 100X oil objectives (NA 1.45). Raw.ims files were processed using Fiji (ImageJ2, v2.14.0), where maximum intensity projections were produced from Z-stacks. The images from *A. franciscana* heads as well as the MOF immunofluorescence stainings were acquired on a spinning disc confocal microscope, VisiScope 5 Elements (Visitron Systems GmbH), which is based on a Ti-2E (Nikon) stand and equipped with a spinning disc unit (CSU-W1, 50 µm pinhole; Yokogawa). The set-up was controlled by the VisiView 5.0 software and images were acquired with a 60x/1.2NA water immersion (CFI Plan Apo VC60x WI) and a sCMOS camera (BSI; Photometrics). The image processing was conducted the same as for the BC43. Fiji (ImageJ2, v2.14.0) was used to analyze foci counts and distances from Z-stack projections. Signal intensities of DAPI, Lamin A/C and H3K9me3 stainings were also quantified in Z-projected images using Fiji. Nuclear size was inferred by manually measuring the area of Z-projected nuclei with the Fiji ROI Manager.

### Statistical testing and data analyses

All plots and statistical analyses were generated in RStudio 2024.04.1 + 748 with R version 4.4.0 (2024-04-24).

### Phylogenetic analysis

The species phylogeny was generated on https://timetree.org/ and visualized on https://itol.embl.de/.

### Supporting information

**S1 Fig. Tissue-specific distribution of H4K16ac in arthropods.** Immunofluorescence staining of H4K16ac in the indicated species with H4K16ac in yellow and DAPI in blue. All images represent maximum intensity projections of a Z-stack. (A) Homogametic sex of indicated species; the heterogametic sex that was stained/acquired as part of the same experiments are presented in Fig 1C. (B) Boxplot showing the *L. decemlineata* DC status from RNA-seq [86] in different tissues by comparing the overall expression in log10(TPM) on X and autosomes in males and females, respectively. Genes with <10 TPM were considered not expressed and thus removed from the analysis. *p*-values were calculated using a two-sided Wilcoxon rank-sum test. (C) Male *A. franciscana* epithelial cells of the abdomen. (D) Head of female *A. franciscana*, showing an overview (left), antennae (middle), and eyestalk (right). The areas highlighted with dashed squares are magnified in the bottom right panel of each 4-square assembly.
(TIF)

**S2 Fig. H4K16ac profiles in arthropod genomes.** (A) Dot plots showing normalized coverage of H4K16ac (CUT&Tag or ChIP-seq, depending on species) in females. Shown is one representative replicate, with chromosomes segmented into 5 kb bins. The *x*-axis represents the relative position of each bin along the chromosome, scaled from 1 (chromosome start) to 100 (chromosome end). Similarly, RNA polymerase II coverage is shown for *Bacillus grandii* males (X0). Pictograms are from phylopic.org. (B) Heatmaps of normalized CUT&Tag coverage in *Bacillus grandii* for RNA polymerase II (left) and H4K16ac (right). Signals are plotted across gene bodies from transcription start site (TSS) to transcription end site (TES) with ±1 kb flanking regions. Pictograms are from phylopic.org. (C) Box plots showing the normalized H4K16ac CUT&Tag coverage of a single representative replicate for each chromosome, segmented into 10 kb bins, in females (grey) and males (white). The experiment was conducted from *n* = 3 biological replicates of each sex. (D) Normalized CUT&Tag enrichment for H4K16ac in comparison with IgG control in one representative female (left) and male (right) sample, shown as a metaplot depicting median enrichment across different gene groups. Gene bodies are scaled to 5 kb between the transcription start site (TSS) and transcription end site (TES), with an additional 1 kb extension upstream and downstream for visualization. Genes were classified as expressed based on a cutoff of TPM ≥ 10. H4K16ac CUT&Tag was independently conducted from *n*=3 biological replicates with similar results (S1 Data for details). The experiment shown here was conducted from frozen *Artemia* tissue. An independent experiment with freshly processed tissue is presented in Fig 3. Both experiments yield similar conclusions, with slightly improved signal-to-noise observed in the data from fresh tissue. (E) log2FC of H4K16ac levels between females and males across chromosome 4 (chosen as one representative autosomal example).
(TIF)

**S3 Fig. Sex-specific expression of MSL-MOF complex and other acetylation factors across arthropods.** (A) Schematic representation of MSL complex architecture (top) and immunofluorescence staining of MOF (purple) and DAPI (blue) in male and female *D.melanogaster*. (B) Heatmap showing the normalized gene expression of MSL complex members as well as selected HAT and HDAC genes for *D. melanogaster* [57], *A. franciscana* [14], *A. gambiae* [23], and *L. decemlineata* [86]. An additional heatmap displays expression of HuR orthologs for *A. franciscana* [14]. Rows represent biological replicates and columns the different genes, respectively. Pictograms are from phylopic.org. (C) Sashimi plots

showing the splicing patterns of MSL complex members in adult *A. franciscana* individuals of both females and males, generated using IGV. The *y*-axis represents the read coverage from the BAM file (not normalized), and the arcs below indicate the reads that span the exon-exon junctions as lines.
(TIF)

**S4 Fig. Conservation of MSL complex members between *A. franciscana* and *D. melanogaster*.** (A) Structural similarity alignment of MOF orthologous protein sequences from *A. franciscana* and *D.melanogaster*, with the chromobarrel and HAT domains highlighted in a box. Structural similarities were assessed using FoldMason and are presented as Local Distance Difference Test (LDDT) scores. (B) as in (A), but for MSL1 orthologous protein sequences, highlighting the coiled-coil and PEHE domains. (C) as in (A), but for MSL2 orthologous protein sequences, highlighting the RING and CXC domains. (D) as in (A), but for MSL3/MRG15 orthologous protein sequences, highlighting the chromo and MRG domains.
(TIF)

**S5 Fig. Gene expression in embryonic development by low-input RNA-seq.** (A) Hierarchical clustering of embryonic low-input RNA-seq samples was performed using Euclidean distance calculations based on rlog-transformed counts in DESeq2. A distance matrix was computed from the rlog-transformed expression values, and a heatmap was generated using the heatmap.2 function from the gplots package in R. (B) MA plots showing differentially expressed genes obtained by DEseq2 between early embryonic stages (1–2 dpf) and later embryonic stages (4 dpf). The plot shows mean normalized counts (*x*-axis) against log2FC (*y*-axis), with significantly changing genes FDR < 0.05 colored in blue. (C) Dot plot showing gene expression values as Log10(TPM + 1) for selected genes in the two stages. Circles (males) or triangles (females) show the values of the individual replicates. (D) Dot plots showing allelic balance (*y*-axis) obtained by Free-Bayes of exonic SNPs along their position on the Z chromosome (*x*-axis). SNPs were filtered for locating in exons, mapping quality >20 and minimum coverage >10. The plots show the results from one adult male (top) and one adult female (bottom) dataset, with the SNPs residing in the Z(diff) region colored in green. (E) as in (D), Violin plots displaying the distribution of allele frequencies of each exonic SNP, comparing the pseudoautosomal region (PAR) and the differentiated region of the Z chromosome in males (top) and females (bottom). (F) as in (D), bar plots comparing the number of biallelic (allelic balance > 0) and monoallelic (allelic balance = 0) SNPs in the PAR and the differentiated region of the Z chromosome. (G) Bar plots showing the mean RNA levels ($n = 3$) of histone acetyltransferase (*CBP/nej, GCN5, TIP60*) and the histone deacetylase (*SIRT1*) genes along embryogenesis obtained by RNA-seq in males and females. The overlaid dots represent the individual Log10(TPM + 1) expression values of a given biological replicate. (H) Manhattan plot of differential transcript expression between male and female *A, franciscana* embryo samples ($n = 6$ per sex; both timepoints were combined for increased statistical power). Each dot represents a 50 bp genomic bin, plotted as -log10(*p*-value) across the 21 chromosomes. Genomic positions are indicated for the most significant hits, and the Z chromosome is highlighted with a green box.
(TIF)

**S6 Fig. H4K16ac levels and gene expression during aging.** (A) Normalized H4K16ac CUT&Tag levels on expressed genes (TPM ≥ 10) in males and females, shown as a violin plot with an overlaid boxplot for each chromosomal region (autosomes, PAR, Z(diff)). H4K16ac enrichment levels for each replicate and sex from TSS-1kb until TES were calculated using deepTools multiBigWigSummary from the normalized signals in counts per million. (B) PCA of H4K16ac levels at expressed genes (TPM ≥ 10) of the differentiated Z chromosome in old (75 days) female and male samples, based on CUT&Tag profiling. Each point represents a single individual/ biological replicate. (C) Dot plot showing the Pearson correlation of mean gene expression with age (*y*-axis) along with the corresponding mean expression value (*x*-axis) for each expressed *Drosophila* gene in females (left plot) and males (right plot). The black horizontal line at 0 marks no decline across aging. The data is a single cell RNA-seq dataset from *Drosophila* brain [70] where 8 time-points

 

until 50 days adult age are sampled. The mean expression across all cells was calculated based on the absolute expression values in each cell (UMI counts). Genes with 0 mean expression in more than half of the sampled time-points were removed. *sta* is an exceptional gene with increased stability upon aging (positive Pearson correlation) and is labelled as a control. MSL and histone acetylation factors are also labelled and decay upon aging (negative Pearson correlation). X chromosomal genes are shown in blue. The dashed blue and black lines represent a log linear regression for X and autosomal genes. (D) Line plot showing how log10(mean expression) of selected histone acetylation factors changes in different adult ages of *Drosophila*, with females shown as dashed and males as solid lines, respectively. Single-cell RNA-seq dataset from [70] as in (C).
(TIF)

**S7 Fig. Lifespan assays and Sirtuin homology.** (A) Line plot showing the mean survival probability with shaded confidence intervals of male and female *Artemia* when they are reared (left:) in a mixed-sex culture or (right:) in separate cultures (no mating). Each replicate culture reflects 50 adults of each genotype seeded. The *x*-axis corresponds to the days after hatching. The experiment was started by culturing the same number of adult stage males and females (25 days) as before the sexes cannot be visually distinguished. The experiment was independently conducted three times, one more replicate is shown in Fig 6A. (B) Sequence alignment of the catalytic domain of *A. franciscana* SIRT1 and *H. sapiens* SIRT1 protein sequences (C) AlphaFold-based 3D structural similarity comparing the catalytic domain of *A. franciscana* SIRT1 protein (jg8800) and human SIRT1. (D) as in (B), but for the full-length SIRT1 protein sequences.
(TIF)

**S1 Data. Mapping Statistics and Details of RNA-seq and CUT&Tag experiments.**
(XLSX)

**S2 Data. Haplotype calling results and expressed SNP analysis in embryos.**
(PDF)

**S3 Data. Genome browser snapshots and BLAST searches of non-coding RNA candidates and IRFinder hits.**
(PDF)

**S4 Data. Results of IRFinder analysis of sex-biased intron retention.**
(XLSX)

## Acknowledgments

We thank Beatriz Vicoso for generously sharing data, engaging in valuable discussions, and providing access to unpublished results. Additionally, we express our gratitude to Francisco Hontoria for kindly teaching us the techniques for rearing *Artemia* and Katharina Papsdorf for critical reading of the original manuscript draft. The *Drosophila* MOF antibody [102] has been kindly obtained from Prof. Peter Becker. Dr. Fridolin Kielisch (IMB bioinformatics core facility; statistical analyses), the IMB protein production core facility (production of enzymes used in this work), the IMB Microscopy and Histology Core Facility, Prof. Marion Silies (equipment and resources for *Drosophila* rearing) and Roxanne Fraser (assisted with termite tissues) are kindly acknowledged. Insect tissues have been kindly provided by Dr. Eric Marois (*A. gambiae;* University of Strasbourg, France), Prof. Shuqing Xu and Léonore Wilhelm (*L. decemlineata*; Johannes-Gutenberg-University Mainz, Germany), Dr. Ines Drinnenberg (*P. interpunctella*, Institut Curie Paris, France). All lab members of the gene dosage lab are acknowledged for their inputs along the project, especially Iona Berger-hoff for assisting with fruit fly rearing. The IMB Genomics Core Facility, Microscopy Core Facility and the use of the NextSeq2000 and Spinning Disk Confocal System (VisiScope, 5-Elements, funded by the DFG - INST 247/912-1FUGG) are gratefully acknowledged.

## Declaration of generative AI and AI-assisted technologies

The authors used ChatGPT along the editing process of the original manuscript draft in order to improve the grammar and clarity of the language of this manuscript. All AI-edited content was reviewed and edited as needed. The authors take full responsibility for the content of the published article.

## Author contributions

**Conceptualization:** Claudia Isabelle Keller Valsecchi.

**Data curation:** Frederic Zimmer, Frank Rühle, Claudia Isabelle Keller Valsecchi.

**Formal analysis:** Frederic Zimmer, Qiaowei Pan, Claudia Isabelle Keller Valsecchi.

**Funding acquisition:** M. Felicia Basilicata, Claudia Isabelle Keller Valsecchi.

**Investigation:** Frederic Zimmer, Annika Maria Fox, Qiaowei Pan, Frank Rühle, Peter Andersen, Claudia Isabelle Keller Valsecchi.

**Methodology:** Frederic Zimmer, Qiaowei Pan, Claudia Isabelle Keller Valsecchi.

**Project administration:** Claudia Isabelle Keller Valsecchi.

**Resources:** Peter Andersen, Ann-Kathrin Huylmans, Tanja Schwander, M. Felicia Basilicata, Claudia Isabelle Keller Valsecchi.

**Software:** Qiaowei Pan, Frank Rühle.

**Supervision:** M. Felicia Basilicata, Claudia Isabelle Keller Valsecchi.

**Validation:** Frederic Zimmer, Claudia Isabelle Keller Valsecchi.

**Visualization:** Frederic Zimmer, Claudia Isabelle Keller Valsecchi.

**Writing – original draft:** Frederic Zimmer, Claudia Isabelle Keller Valsecchi.

**Writing – review & editing:** Frederic Zimmer, Annika Maria Fox, Qiaowei Pan, Frank Rühle, Peter Andersen, Ann-Kathrin Huylmans, Tanja Schwander, M. Felicia Basilicata, Claudia Isabelle Keller Valsecchi.

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
