## [Decision Letter · Decision Letter 0]

21 May 2025

PGENETICS-D-25-00319

Convergent Evolution of H4K16ac-mediated Dosage Compensation Shapes Sex-dependent Lifespan in a ZW Species.

PLOS Genetics

Dear Dr. Keller Valsecchi,

Thank you for submitting your manuscript to PLOS Genetics. After careful consideration, we feel that it has merit but does not fully meet PLOS Genetics's publication criteria as it currently stands. Therefore, we invite you to submit a revised version of the manuscript that addresses the points raised during the review process. Most important were the critiques from two reviewers that the results as they stand do not include enough functional validation to support the claim of a causal role for dosage compensation breakdown in aging or lifespan differences between sexes in Artemia. I agree with the reviewers' suggestions that you either make additions that strengthen the aging part of the story or put it aside and strengthen the functional inferences from the work focused on H4K16 acetylation as a mediator of dosage compensation.

Please submit your revised manuscript within 60 days Jul 20 2025 11:59PM. If you will need more time than this to complete your revisions, please reply to this message or contact the journal office at plosgenetics@plos.org. Please include the following items when submitting your revised manuscript:

We look forward to receiving your revised manuscript.

Kind regards,

Rachel Brem

Academic Editor

PLOS Genetics

Paula Cohen

Section Editor

PLOS Genetics

Aimée Dudley

Editor-in-Chief

PLOS Genetics

Anne Goriely

Editor-in-Chief

PLOS Genetics

**Journal Requirements:**

1) Please provide an Author Summary. This should appear in your manuscript between the Abstract (if applicable) and the Introduction, and should be 150-200 words long. The aim should be to make your findings accessible to a wide audience that includes both scientists and non-scientists. Sample summaries can be found on our website under Submission Guidelines:

https://journals.plos.org/plosgenetics/s/submission-guidelines#loc-parts-of-a-submission

2) We noticed that you used the phrase 'data not shown' in the manuscript. We do not allow these references, as the PLOS data access policy requires that all data be either published with the manuscript or made available in a publicly accessible database. Please amend the supplementary material to include the referenced data or remove the references.

- ® on pages: 20, and 23.

Potential Copyright Issues:

i) Figures 1A, 2A, 3A, and S3. Please confirm whether you drew the images / clip-art within the figure panels by hand. If you did not draw the images, please provide (a) a link to the source of the images or icons and their license / terms of use; or (b) written permission from the copyright holder to publish the images or icons under our CC BY 4.0 license. Alternatively, you may replace the images with open source alternatives. See these open source resources you may use to replace images / clip-art:

5) Please ensure that the funders match between the Financial Disclosure field and the Funding Information tab in the online submission form. Both locations should list the same funders, grant numbers, and recipients in the same order.

**Reviewers' comments:**

Reviewer's Responses to Questions

Reviewer #1: Overall comments

Dosage compensation/dosage balance in the evolution of chromosomal sex determination is a fascinating problem, because it seems that rapid evolution of sex determination systems goes alongside enormously diverse ways in which dosage compensation happens, from either no compensation at all through to chromosomal upregulation, chromosomal downregulation and/or inactivation. The ‘rules’ of this, if there are any, are completely unknown. The only way we can get enough information to even begin to speculate is through studying many different sex chromosomal systems to see whether each system is unique or whether there are any commonalities. In this regard, the current manuscript is a really important study because it considers dosage compensation in a ZW system (the crustacean Artemia), which are currently understudied at the molecular level due to lack of appropriate model systems. The manuscript thus tackles an important question. Moreover, it is largely novel- it reports chromatin-based dosage compensation through H4K16Ac, which is similar (although not identical) to the Drosophila dosage compensation mechanism, even though Drosophila is XY rather than ZW. Most of the results are presented quite well and some key questions are answered, such as the differences between pseudo-autosomal regions that are still recombining and more divergent regions; other questions, such as exactly how the H4K16Ac is recruited will hopefully open up new avenues of research for the future.

Despite these major strengths, the paper has a few weaknesses. The first is that it feels as though two rather distinct papers, one focussed on dosage compensation and one focussed on ageing, have been cobbled together. Whilst connected conceptually, the problem with this is that neither of the two areas has been explored in sufficient detail. It seems that it would have been better if one area had been more fully studied, with two separate papers eventually produced. In my opinion, the dosage compensation aspect is more thorough at present, but even this is missing some key aspects of data analysis, and would benefit from some further experiments to better characterise the system. The effect of H4K16Ac on ageing is weaker and needs more work to be convincing. I think that if the authors want to keep both of these in the same paper, they need to do more on both aspects. In the specific experimental/analytic comments below I have some suggestions for what I think needs doing to make the paper stronger.

Major points

1) The authors identify the Artemia system as one where K16Ac is important on the basis of chromosome staining followed by ChIP. This is convincing. However, this approach cannot rule out that other species which do not show a K16Ac territory do not have K16Ac sex-chromosome dosage compensation- it is not necessarily the case that a territory is always present whenever K16Ac enrichment on the sex chromosome is present. It would be good if the authors, as a control, could do ChIP on a system that they would predict would NOT have sex chromosome K16Ac, and confirm their prediction. This would justify the screening approach as a way to identify potential K16Ac systems.

2) The K16Ac enrichment does appear clear from the ChIP. However, there is an important concern- it is not clear from the Figure legend how the normalization in Figure 2’s genome brower snapshots was carried out. Moreover, the scale shown is different between the two genotypes. Crucially, it is technically possible that the Cut and Run profile could be an artefact of higher accessibility. The authors presumably have normalized to IgG controls in the snapshot, but it would be preferable to actually show the IgG traces too, to confirm that they are not different between the two genotypes. The normalization relies on linearity, which may not be a valid assumption in this case, so if the IgG is much higher in one genotype than another, it could indicate that the K16Ac enrichment is artefactual.

3) A very important point is that the primacy of K16Ac in the dosage compensation cannot be established without understanding the mechanism of recruitment. The authors do not find any evidence of sex-specific K16Ac complex expression, and cannot show increased recruitment of the Acetylation complex. Histone modifications obviously correlate, so some other mark may be the driver of the K16Ac. An alternative, perhaps less likely but very interesting possibility is that there is a two-tier system happening here: it could be that there is downregulation of the entire chromosome by increased repressive histone marks, but then in the heterogametic sex upregulation by K16Ac occurs on top. In an ideal world it would be nice if the authors could do four further ChIPs which would give a bit more insight by covering major marks implicated before in dosage compensation in other systems:

H3K4me3

H3K9me2

H3K27me3

H3K36me3

Whilst this is a fair amount of work, I think it would be important if the authors wish to argue that this is an example of a K16Ac dominated dosage compensation system, rather than something more complex. For example, if there were similar expansion of H3K4me3 beyond promoters, this might suggest that it too is important in the dosage compensation (this doesn’t happen in Drosophila); if there is also enrichment of silencing marks either in the homogametic sex or in both sexes that might suggest a more complex, two-tier system is in operation.

4) The ageing aspect of the paper is very interesting, but at present over-interpreted. The key problem is that the experiment with nicotinamide is not strong. There is no evidence directly presented that nicotinamide changes H4K16Ac levels. Without this it’s impossible to be certain that this inhibitor’s effect on ageing is anything to do with SIRT1 or sex chromosome dosage at all. If they want to argue this then they need to do ChIP after nicotinamide treatment, and also show that changes in gene expression correlate strongly with areas where the nicotinamide treatment shows the largest effects.

Minor points

The manuscript is generally well written but some of the figure callouts need checking- I noticed that Fig 2 is referred to as Fig 1 in some parts of the manuscript.

Also, the use of the word ‘vulnerability’ in the abstract is not very clear- it would be better to rephrase the sentence as:

‘They highlight the potential role of sex-specific histone acetylation in driving male-female differences in ageing’ or something similar.

Reviewer #2: The work by Frederic Zimmer and colleagues reports the identification of the dosage compensation (DC) mechanism acting on the sex chromosomes of the crustacean Artemia franciscana, a female-heterogametic (ZW) species. Sex chromosomes often show marked structural heteromorphism, leading to a fundamental imbalance in the dosage of sex-linked genes—one sex carries two copies of the chromosome, while the other has only one. To resolve this imbalance and ensure equal gene expression between the sexes, many organisms have evolved chromosome-wide, epigenetic-based dosage compensation mechanisms. However, the molecular machinery underpinning DC is well-characterized in only a few model systems.

In the first part of the paper, Zimmer and colleagues report the remarkable discovery that Artemia has independently evolved an H4K16ac-based mechanism of dosage compensation, convergent with that seen in Drosophila. This section is of very high quality—the data are compelling, the analyses are thorough, and the conclusions are well supported. The findings represent a significant and high-impact contribution to the field of sex chromosome biology.

The second part of the manuscript explores the intriguing hypothesis that dosage compensation mechanisms may create sex-specific epigenetic vulnerabilities later in life, potentially contributing to sex differences in longevity. This idea is novel and certainly worth exploring. However, the current evidence is limited, lacks functional validation, and suffers from several important caveats. In fact, one could reasonably draw the opposite conclusion from the data: that dosage compensation is robust and remains stable with age, unlike other parts of the genome.

For instance, the choice of 75 days as the “old” time point may be too late to detect causal aging processes, as animals at this stage are likely to be terminal. H4K16ac levels on the Z chromosome do not exhibit significant age-related changes (Fig. 4C), and age-associated gene expression changes are not restricted to the Z chromosome (Fig. 4E). Interestingly, the Z(diff) genes examined show a slight increase in expression, which contradicts expectations under a model of dosage compensation erosion—reduced H4K16ac would typically be associated with decreased expression. Furthermore, nicotinamide treatment alters gene expression more broadly on the PAR and autosomes than on the Z(diff) region in females (Fig. 5D), suggesting that the Z chromosome is not uniquely vulnerable. Taken together, these findings do not support the conclusion that Z-linked dosage compensation creates a specific epigenetic vulnerability during aging.

In summary, this study presents highly valuable and novel data that merit publication. However, several of the claims of the second part of the paper—particularly those made in the title, abstract, and later parts of the manuscript—should be revised to reflect the preliminary nature of the evidence.

Overall, I strongly support the publication of this work in PLOS Genetics, pending these revisions.

Please find below my detailed comments and suggestions:

Major comments:

Line 242 – Paragraph: Alterations of H4K16ac on the Z chromosome of old female

The rationale for selecting 75 days as the "old" time point is unclear and appears unjustified. At this stage, over 95% of the population has already died, with a median lifespan of 45 days. This time point is therefore far too late to capture biological events that drive the ageing process. Instead, it is likely to reflect late-stage, indirect, or downstream consequences of ageing. To properly assess the dynamics of H4K16ac acetylation on the Z chromosome during ageing, a time-course analysis using immunofluorescence staining should be conducted at intermediary ages. Suggested time points could include 30, 40, 50, and 60 days. For the findings to be relevant to the ageing process, any observed changes should occur before the median lifespan.

Line 251 – “there was a declustering indicated by a significant increase in the mean and maximum distance between H4K16ac foci in old females.”, and Line 253-5 – “This revealed that the average levels of H4K16ac on genes on the Z(diff) region and autosomes remain similar when individuals age.”

I find it difficult to reconcile the CUT&Tag data, which show no temporal changes in H4K16ac levels, with the immunofluorescence results. Could the authors elaborate on the biological significance of the observed “declustering”? The CUT&Tag results suggest that there is no de novo acetylation of genomic regions outside the Z chromosome. What, then, do these newly observed acetylation clusters correspond to? Could they reflect a deterioration in the spatial organization of the Z chromosome territory within the nucleus, occurring independently of changes in acetylation levels? If so, is this effect specific to the Z chromosome, or does it point to a broader disruption of nuclear architecture? To address this, could the authors assess nuclear organization more broadly using markers for other nuclear compartments (e.g., the nuclear lamina or nucleolus), or ideally, perform FISH targeting both the Z chromosome and representative autosomes?

Line 317 – “more autosomal and PAR genes were affected in females.”, and lines 329-331 – “Taken together, interfering with the acetylation balance affects gene expression on sex chromosomes differently in the two sexes and this is associated with male-female lifespan differences.”

These are overstatements that need to be corrected. First, no direct link between H4K16 acetylation and nicotinamide treatment is presented. The authors should assess H4K16ac levels in males and females following nicotinamide exposure. Even if differences were observed, they would represent a correlation, not evidence of a causal relationship. Furthermore, nicotinamide treatment affects a greater number of autosomal and PAR genes, indicating that most of its effects are not mediated by specific deacetylation of the female Z chromosome and are likely unrelated. Finally, nicotinamide is clearly toxic to both sexes. Based on the current data, no conclusions can be drawn regarding the involvement of dosage compensation degradation in the regulation of sex differences in longevity.

Minor comments:

Lines 37-9 – “along with transcriptome analysis in the "model" ZW organism, the chicken, initially led to the assumption that DC would be entirely absent in ZW systems [8] or, if present, would act only on few dosage-sensitive genes”

An important reference is missing and should be discussed here: https://doi.org/10.1101/2024.03.06.581755. In this recent study, the Kaessmann lab identified a molecular mechanism of dosage compensation in chicken. They functionally characterized a Z chromosome-linked microRNA, miR-2954, which exhibits strong male-biased expression. Knockout of miR-2954 led to early embryonic lethality in homozygous knockout males only, driven by the specific upregulation of dosage-sensitive Z-linked target genes. Moreover, they showed that miR-2954 represses the expression of approximately 50% of all 375 Z-linked genes, highlighting its role in mediating Z chromosome dosage compensation.

Lines 52-4 – “While mammals and C.elegans use repressive chromatin modifications in XX individuals,”

This represents only the second step in the dosage compensation mechanism. It is important to mention that a first step involves the upregulation of X chromosome activity to balance gene expression from the single X chromosome in males with that of the two sets of autosomes. This phenomenon of X-chromosome upregulation has been well documented in recent studies, including work in C. elegans (DOI: 10.1093/gbe/evaf061) and in certain eutherian mammals (DOI: 10.1126/science.abd8887).

Fig. 2C – The figure legend incorrectly states "male (left) and female (right)"; this should be corrected to "female (left) and male (right)."

Line 173 – Paragraph: Absence of sex-specific patterns of MOF-MSL complex in Artemia.

The data presented are clear and compelling; however, some interpretations appear somewhat overstated. The absence of sex-specific transcription does not rule out other layers of sex-specific regulation. For instance, in Drosophila, Msl-2 protein is expressed exclusively in XY cells due to post-transcriptional regulation, including sex-specific splicing and translational repression in the cytoplasm. Therefore, a sex-specific pattern of MOF activity in Artemia could still occur in the absence of transcriptional sex bias. Some of the claims should remain framed as hypotheses—for example, line 240: “This provides further support for the idea that H4K16ac is deposited by an MSL-independent mechanism” should be presented more cautiously.

Along the same lines, the authors should expand the discussion of their transcriptional profiling results, particularly regarding any protein-coding genes or non-coding RNAs with sex-specific expression. For instance, what is the expression status of factors from the Human antigen R (HuR/RRM) family? This is relevant given that the Vicoso lab recently identified an AT-rich motif highly enriched in the S0 region of the Z chromosome that may be involved in targeting the dosage compensation machinery. This motif was previously characterized as a binding site for HuR proteins (https://doi.org/10.1093/molbev/msaf085).

Fig. 4 and Fig. 5 – The figure labels appear to be inverted and should be corrected to match the appropriate content; what is currently labeled as Fig. 4 corresponds to Fig. 5, and vice versa.

Fig. 4C – The legend states that the three replicates are shown as overlays with different shades, but these are not visible in the figure.

Line 257 – “In old females, H4K16ac levels on the female Z(diff) became more variable.”, and Line 263 – “The increased variability in H4K16ac was accompanied by gene expression changes”.

I find the proposed model difficult to interpret. H4K16ac appears to become slightly more variable specifically on the Z(diff), yet the observed changes in gene expression are not restricted to this region. Moreover, all Z(diff) genes examined show a slight increase in expression, which is counterintuitive given that a decrease in H4K16ac would typically be expected to reduce gene expression. As previously mentioned, the chosen time point is extremely late in the lifespan. Based on the presented data—particularly the lack of significant changes in the acetylation profile—I would draw the opposite conclusion from the authors: that dosage compensation remains stable and is not notably affected by aging.

Line 280 - “Altogether, our findings are consistent with the idea of inappropriate DC becoming a vulnerability when heterogametic individuals get older. Accordingly, perturbations in H4K16ac would affect the X-linked genes in Drosophila males, while they impact Z-linked genes in Artemia females.”

The authors should revise this paragraph and temper their interpretation to more accurately reflect the data. There is no direct evidence supporting this claim, and the data appear to suggest the opposite. PAR gene expression becomes also more variable with age (Fig. 4E), and nicotinamide treatment affects gene expression more broadly on the PAR and autosomes than on the Z(diff) region in females (Fig. 5D). In addition, H4K16ac levels on the Z chromosome do not show significant age-related changes (Fig. 4C). Taken together, these findings do not support the idea that dosage compensation of the Z chromosome creates a specific vulnerability in aged individuals. On the contrary, the data suggest that other genomic regions—particularly the PAR and autosomes—are more susceptible to aging-associated changes.

Line 293 – “This revealed that several factors change expression but also show increased variability in old females, likely perturbing histone acetylation homeostasis.”

Why would gene expression variability be higher in older females? Could the authors clarify whether the tested genes are Z-linked? If these effects were due to a degradation of dosage compensation, the findings appear counterintuitive—since a decrease in H4K16ac is generally associated with reduced, not increased, gene expression. Furthermore, the magnitude of the observed changes is relatively small, which raises questions about their potential biological relevance.

Line 307-9 – “Interestingly, the sex-specific differences in lifespan observed under control conditions were alleviated in nicotinamide, indicating that the susceptibility to SIRT1 inhibition differs between males and females.”

This sentence needs clarification, as it is quite misleading. The effect of nicotinamide treatment on longevity is extremely strong and appears more consistent with toxicity than with regulated lifespan modulation. In males, the median adult lifespan drops from approximately 18 days to less than 5 days—a reduction of about 75%. Moreover, all individuals, regardless of sex, die rapidly. At this concentration, nicotinamide is clearly toxic and lethal to Artemia.

Line 310 – “To elucidate how the nicotinamide treatment alters gene expression, we performed RNA-seq from n=4 adult age-matched males and females subjected to nicotinamide for 30 days from hatching onwards in comparison with untreated controls.”

The rationale for selecting a 30-day treatment time point is unclear and should be explicitly justified. Given that the median adult lifespan under nicotinamide treatment is approximately 5 days (Fig. 5C), sampling at 30 days raises concerns that the data may reflect dying or already dead animals, rather than specific biological responses.

Line 378 – “By revealing a link between misregulated DC and ageing, our data can further provide one mechanistic explanation for reduced longevity of the heterogametic sex in both XY and ZW species.”

These are overstatements that should be corrected. H4K16ac levels on the Z chromosome do not show significant age-related changes, and the findings presented are too preliminary to support the claim that Z chromosome dosage compensation creates a specific vulnerability in aged individuals.

Line 383 – “Unexpectedly, roX1 lncRNA has been recently uncovered as a fruit fly ageing biomarker and interfering with roX1 function partially alleviates sex-specific differences in lifespan [56]. Although the effect of MSL-H4K16ac in the Drosophila context (roX1 mutant) remains to be investigated, one possibility is a link to DC.”

Not exactly. The rox1[SMC17A] mutant specifically affects female lifespan and has no impact in males, where MSL-mediated dosage compensation occurs. By definition, the novel function of rox1 described here is therefore independent of the dosage compensation pathway.

Line 390 – “Mechanistically, this could be linked to mechanisms like epigenetic noise and transcriptional instability [74], observed by us here in Artemia but also found in other systems like mammals. Increased cell to-cell transcriptional variability occurs for example in the ageing mammalian immune system [75]. In mammals, genes that escape X inactivation - resulting in biallelic expression in XX females but single-copy expression in XY males - may play a crucial role that is worthwhile investigating in the future [60,76].”

The proposed model is unclear. If aging in XX females were driven by erosion of X inactivation leading to biallelic expression, one would expect this to result in reduced lifespan. However, XX females generally live longer than males. If dosage compensation indeed created a vulnerability in mammals, this pattern would be reversed. The model, as currently presented, does not align with known sex differences in lifespan.

Title – “Convergent Evolution of H4K16ac-mediated Dosage Compensation Shapes Sex-dependent Lifespan in a ZW Species.”

The second part of the title, “shapes sex-dependent lifespan in a ZW species”, should be removed or significantly revised, as the current data do not provide sufficient evidence to support such a strong and causal claim.

Abstract – The following three sentences should be substantially revised to more accurately reflect the data:

-“H4K16ac-mediated DC is established during embryogenesis, sustained until adulthood, but then becomes reconfigured and variable in aged females.”

This statement is not supported by the data, as the authors themselves report no significant age-related changes in H4K16ac levels on the Z chromosome (Fig. 4C).

-“Interfering with acetylation alleviates male-female differences in lifespan.”

This is misleading. The effect of nicotinamide treatment is extremely severe and suggests general toxicity rather than modulation of sex-specific lifespan. In males, the median lifespan drops from ~18 days to <5 days—a ~75% reduction—and both sexes die rapidly. At the tested concentration, nicotinamide appears to be broadly toxic and lethal to Artemia, not a targeted regulator of sex-specific longevity.

-“They highlight histone acetylation as a sex-specific vulnerability linked to the chromatin landscape of sex chromosomes and its potential role in driving male-female differences.”

This claim overstates the findings. The presented data are preliminary and do not support the conclusion that dosage compensation of the Z chromosome constitutes a sex-specific epigenetic vulnerability in aging.

Reviewer #3: This is a very nice contribution to our understanding of the evolution of dosage compensation. The conclusions are mostly well supported. In particular, the data presented in Figure 2 and 3 especially. I have some minor comments, but I do have one major concern that should be addressed either with new experiments or a very clear statement hedging their conclusions.

Major point: One conclusion from figure 1 might be that Plodia, Bacillus, Anopheles, Leptinofarsa and Reticulitermes and Triops do not achieve dosage compensation on the Z or W in the heterogametic sex through H4K16ac. However, that specific conclusion is not supported because the IF they do doesn't provide information about the sex chromosomes. The sex chromosome may dispersed within the nuclei, still significantly marked with H4K16ac, but simply not partitioned in a distinct territory. The specific conclusion that can be made is that there is no distinct H4K16ac TERRITORY in the heterogametic sex. To conclude that there is no DC by way of H4K16ac in the heterogametic sex, they would have to either mark the sex chromosome with FISH, or do chromatin profiling with ChIP seq or cut and run. As such, they must provide a clear statement indicating that they may only conclude that any possible dosage compensation in these species may still occur through H4K16ac, but that there doesn't appear to be a paritioning as seen in Drosophila, Bradysia and Artemia.

Minor points:

Line 50: Italicize A. franciscana

Line 172: Worth mentioning the RNA-seq result for 1F which supports their conclusion. This doesn't appear mentioned in the text.

How is FTZ dynamically expressed in S5C? I don't see it. Likewise for nAChRB1. Others, perhaps so. Please be more upfront about this. It is hard to know, with the variation presented, if there is some significant difference across stages.

Are 4 and 5 switched with respect to order???

Provide some visual that demonstrates source of variation in 4D. Is this always driven by a single sample? Or, is it variable across samples (ie, in one sample, one gene in z(diff) is quite different from the other two, but, in the same sample comparison, another gene shows a different pattern. Could this be driven by a single old sample? This needs to be clarified.

Overall, I enjoyed this paper and I think it is a nice contribution.

**Have all data underlying the figures and results presented in the manuscript been provided?**

Reviewer #1: Yes

Reviewer #2: Yes

Reviewer #3: Yes

PLOS authors have the option to publish the peer review history of their article (what does this mean? ). If published, this will include your full peer review and any attached files.

**Do you want your identity to be public for this peer review?** For information about this choice, including consent withdrawal, please see our Privacy Policy .

Reviewer #1: No

Reviewer #2: No

Reviewer #3: No

**Figure resubmission:**
---

## [Decision Letter · Decision Letter 1]

28 Sep 2025

Dear Dr Keller Valsecchi,

We are pleased to inform you that your manuscript entitled "Convergent Evolution of H4K16ac-mediated Dosage Compensation in the ZW Species Artemia franciscana." has been editorially accepted for publication in PLOS Genetics. Congratulations!

Yours sincerely,

Rachel Brem

Academic Editor

PLOS Genetics

Paula Cohen

Section Editor

PLOS Genetics

Aimée Dudley

Editor-in-Chief

PLOS Genetics

Anne Goriely

Editor-in-Chief

PLOS Genetics

BlueSky: @plos.bsky.social

Comments from the reviewers (if applicable):

Reviewer's Responses to Questions

**Comments to the Authors:**

Reviewer #1: I thank the authors for their careful and considered responses to my questions from the first round of review, which have satisfied the technical and theoretical concerns that I had very well and strengthened the manuscript considerably. I'm looking forward to seeing this important work published.

Reviewer #2: The authors have thoroughly addressed all of my previous comments, performing numerous additional experiments that have strengthened this updated version of their manuscript. The weaknesses present in the earlier version have been fully resolved with the thoughtful corrections provided. I also support the removal of the Nicotinamide-treatment, which—as the authors rightly noted—raised more questions than it answered.

The analysis of dosage compensation across life stages in Artemia, combined with the many valuable new additions, results in a highly compelling and exciting manuscript that clearly deserves publication in PLOS Genetics. I warmly congratulate the authors on their significant findings.

Reviewer #3: The authors did a nice job following up on my concerns. The data now presented in figure 2 supports the conclusion that H4K16ac enrichment on the differentiated Z is probably convergent with what is seen for the Drosophila X

**Have all data underlying the figures and results presented in the manuscript been provided?**

Reviewer #1: Yes

Reviewer #2: Yes

Reviewer #3: Yes

PLOS authors have the option to publish the peer review history of their article (what does this mean? ). If published, this will include your full peer review and any attached files.

**Do you want your identity to be public for this peer review?** For information about this choice, including consent withdrawal, please see our Privacy Policy .

Reviewer #1: **Yes: ** Peter Sarkies

Reviewer #2: No

Reviewer #3: No

**Data Deposition**

http://datadryad.org/submit?journalID=pgenetics&manu=PGENETICS-D-25-00319R1

**Press Queries**

---

## [Editor Report · Acceptance letter]

PGENETICS-D-25-00319R1

Convergent Evolution of H4K16ac-mediated Dosage Compensation in the ZW Species Artemia franciscana.

Dear Dr Keller Valsecchi,

We are pleased to inform you that your manuscript entitled "Convergent Evolution of H4K16ac-mediated Dosage Compensation in the ZW Species Artemia franciscana." has been formally accepted for publication in PLOS Genetics! Your manuscript is now with our production department and you will be notified of the publication date in due course.

With kind regards,

Anita Estes

PLOS Genetics

On behalf of:
